# Non-genetic differences underlie variability in proliferation among esophageal epithelial clones

**Raúl A. Reyes Hueros[1], Rodrigo A. Gier[2], Sydney M. Shaffer**  [2,3] *

**1** Department of Biochemistry and Molecular Biophysics, University of Pennsylvania, Philadelphia, Pennsylvania, United States of America, **2** Department of Bioengineering, University of Pennsylvania, Philadelphia, Pennsylvania, United States of America, **3** Department of Pathology and Laboratory Medicine, Perelman School of Medicine, University of Pennsylvania, Philadelphia, Pennsylvania, United States of America

* sydshaffer@gmail.com

## Abstract

Individual cells grown in culture exhibit remarkable differences in their growth, with some cells capable of forming large clusters, while others are limited or fail to grow at all. While these differences have been observed across cell lines and human samples, the growth dynamics and associated cell states remain poorly understood. In this study, we performed clonal tracing through imaging and cellular barcoding of an in vitro model of esophageal epithelial cells (EPC2-hTERT). We found that about 10% of clones grow exponentially, while the remaining have cells that become non-proliferative leading to a halt in the growth rate. Using mathematical models, we demonstrate two distinct growth behaviors: exponential and logistic. Further, we discovered that the propensity to grow exponentially is largely heritable through four doublings and that the less proliferative clones can become highly proliferative through increasing plating density. Combining barcoding with single-cell RNA-sequencing (scRNA-seq), we identified the cellular states associated with the highly proliferative clones, which include genes in the WNT and PI3K pathways. Finally, we identified an enrichment of cells resembling the highly proliferative cell state in the proliferating healthy human esophageal epithelium.

## Author summary

Single cells derived from tissues and cell lines can have variable abilities to grow in culture. Understanding the molecular differences that lead to this variability has the potential to reveal regulators of growth and to improve in vitro culture for tissue regeneration. In this study, we investigated differences in growth between esophageal epithelial cells and found that these differences are non-genetic in origin. Through quantitative analysis of clonal growth at different time points, we show that the heterogeneity in growth can be modeled as two separate populations, one that is highly proliferative and grows exponentially, and another that is less proliferative and is able to transition into a non-proliferative state. Tracking clones through cell barcoding with scRNA-seq, we define the molecular

**Data Availability Statement:** The scRNA-seq data generated for this study is publicly available in the Gene Expression Omnibus (GEO) repository under accession number GSE262606 and GSE279374. The sequenced genomic DNA (gDNA) data can be

accessed through FigShare at the following project link: https://figshare.com/projects/Non-genetic_differences_underlie_variability_in_proliferation_among_esophageal_epithelial_clones/196885. All data to reproduce the results in the paper are available on FigShare under the same link. All code used in the analysis and for generating figures is available on GitHub: https://github.com/raanreye/EsophagealCloneVariability. Raw data for the healthy esophageal epithelium samples from other studies were downloaded from Gene Expression Omnibus (GEO), accession numbers: GSE201153 and GSE188955, and from European Genome-phenome Archive (EGA), accession number: EGAD00001010074.

**Funding:** S.M.S. acknowledges support from the NIH Director's Early Independence Award DP5OD028144, a donation to the Institute for Regenerative Medicine at the University of Pennsylvania from Larry and Mickey Magid, and the Institute for Translational Medicine and Therapeutics of the Perelman School of Medicine at the University of Pennsylvania (NIH NCATS UL1TR001878). R.A.R.H. acknowledges support from the NSF Graduate Research Fellowship (DGE-1845298). The funders had no role in study design, data collection and analysis, decision to publish, or preparation of the manuscript.

**Competing interests:** The authors have declared that no competing interests exist.

differences between these populations and their associated signaling pathways. Together, this work highlights the contributions of non-genetic heterogeneity in proliferation, which has applications for tissue regeneration and stem cell biology.

## Introduction

Single cells can have variable efficiencies for growth in culture, particularly when grown alone as individual cells. Within a cell line or human sample, this variability can be seen as some clones are able to grow, forming large clones, while others grow partially or fail entirely [1–4]. These differences are generally hypothesized to reflect differences in differentiation status or stemness of the cells [5]. Understanding the basis for these differences in clones has the potential to improve methods for tissue regeneration and stem cell biology [6].

Human epithelial tissues have been used to study these clonal dynamics. An example of such is the human esophageal epithelium in which the native tissue consists of a basal layer of slowly dividing stem cells which give rise to a highly proliferative suprabasal layer and increasingly differentiated progeny cells above [7–9]. In the esophageal epithelium, progenitor cells can have different capacities for growth, resulting in clones of different sizes contributing unequally to the tissue [10,11], similar to differences in clonal potential seen in vitro [12]. In some instances, enhanced growth can be achieved through genetic mutations that alter the equilibrium between proliferation and differentiation, resulting in the generation of a greater number of proliferative progenitors from a single cell [10,13–17]. Thus, clones that acquire these mutations outcompete other neighboring clones that are less proliferative. However, in other contexts, epithelial clones can have different growth potential in the absence of a genetic driver, potentially reflecting differences in stemness and differentiation [12,18,19].

In other biological systems, populations of cells can take on different phenotypes without genetic differences. These differences can stem from differences in gene expression and the accumulation of epigenetic modifications, such as DNA methylation or histone modifications [20–25]. Examples of non-genetic phenotypes are broad and can include therapy resistance [26–30], cancer cell fitness [31,32], immune signaling [33–35], and circadian period [36]. Notably, these phenotypes, despite their non-genetic origins, are marked by distinct gene expression states and can be passed on through cell division [27,30]. Thus, we wondered whether cell growth differences in culture might also exhibit this limited heritability through cell division and be linked to specific gene expression states.

We first performed a quantitative analysis of the growth of an esophageal epithelial cell line at low density. We found a subset of clones that exhibited significantly higher proliferative rates, referred to here as highly proliferative clones. The remaining clones were also able to grow, but they were less proliferative and varied dramatically in their final size after 11 days. Using mathematical modeling, we found that the growth of the highly proliferative clones can be fitted by an exponential growth model, while the growth of the less proliferative clones can be modeled by the same doubling rate, but through the addition of another parameter that consists of a transition rate to a non-proliferative state at day eight. Further, we found that the highly proliferative state is reversible, suggesting that the proliferative phenotype is non-genetic in its origin. Finally, by pairing DNA barcoding with scRNA-seq, we uncover the distinct gene expression present in the highly proliferative and less proliferative clones indicating differences that extend beyond just the cell cycle.

## Results

### Individual epithelial clones have unique capacities for growth and differentiation

To study single-cell heterogeneity in growth dynamics, we used an in vitro model consisting of esophageal epithelial cells immortalized with telomerase (EPC2-hTERT) [37]. We selected this model to allow us to make quantitative measurements of growth, which are not possible in human tissues. We first seeded individual EPC2-hTERT cells into separate wells of a 96-well plate and allowed them to grow for 14 days. We noted significant heterogeneity in growth during this time period. A small fraction of clones expanded to fill entire wells, while the majority exhibited reduced growth or stopped growing entirely. We quantified the total number of clones that reached confluency in the wells and found that this number was approximately 10% across biological and technical replicates (S1A and S1B Fig and S1 Table).

To increase the number of clones observed, we performed the same experiments in a 6-well plate format, seeding each well with 50 cells per well and then allowed them to grow for 2 to 11 days (Fig 1A). In this experimental design, the cells are sufficiently spread apart that the clones do not come into contact with each other. We fixed a subset of these 6-well plates at discrete time points, including two, five, eight, and 11 days, and then quantified the number of cells in each clone using imaging and counting the nuclei (Fig 1A, left). The sparse initial seeding made it possible to distinguish individual clones as distinct clusters on the plate (Fig 1B). At each time point, we observed a variable range of clone sizes increasing with time of culture (Figs 1C and S2). By day eight, the number of cells per clone ranged from 1 to 2,907 cells. We further confirmed the presence of a subset of highly proliferative clones using live cell imaging (S3E and S3F Fig). However, we noted that the live cell imaging conditions yielded smaller clones across the distribution at all time points.

To determine whether the size differences between clones reflect differences in proliferative cells, we performed immunofluorescence staining. We used Ki-67, a marker of proliferating cells, and phalloidin, which stains F-actin. We found a distinctive elongation and F-actin staining in the cells within the smaller clones, while the cells in the larger clones stained positive for Ki-67 and displayed a more rounded and compact morphology (Fig 1D and 1E). Notably, even among the smaller clones, we still observed a small population of cells staining positive for Ki67, suggesting that while these clones overall grow much less, they can still contain proliferating cells. Thus, we classified clones based on their overall size, with the largest 10% being designated as highly proliferative, and the remaining as less proliferative.

We next wondered whether the highly proliferative clones were a genetically different subset within the EPC2-hTERT line. To test this hypothesis, we isolated single cells from the EPC2-hTERT line using limiting dilutions in a 96-well plate. After 14 days, we identified the most proliferative clones and replated them using a second limiting dilution into another 96-well plate (S1C Fig). If genetic differences in these clones were responsible for the growth, we would expect that all of the subclones derived from the highly proliferative parent clone would grow similarly to the parent. Instead, the growth patterns of these subclones followed a similar growth distribution to the initial population (S1D Fig). We similarly tested replating single cells derived from less proliferative clones, but did not observe any highly proliferative clones (S1D Fig). Given that the highly proliferative state of these clones can be reversed, we concluded that they are not a genetically distinct subset of the EPC2-hTERT cell line.

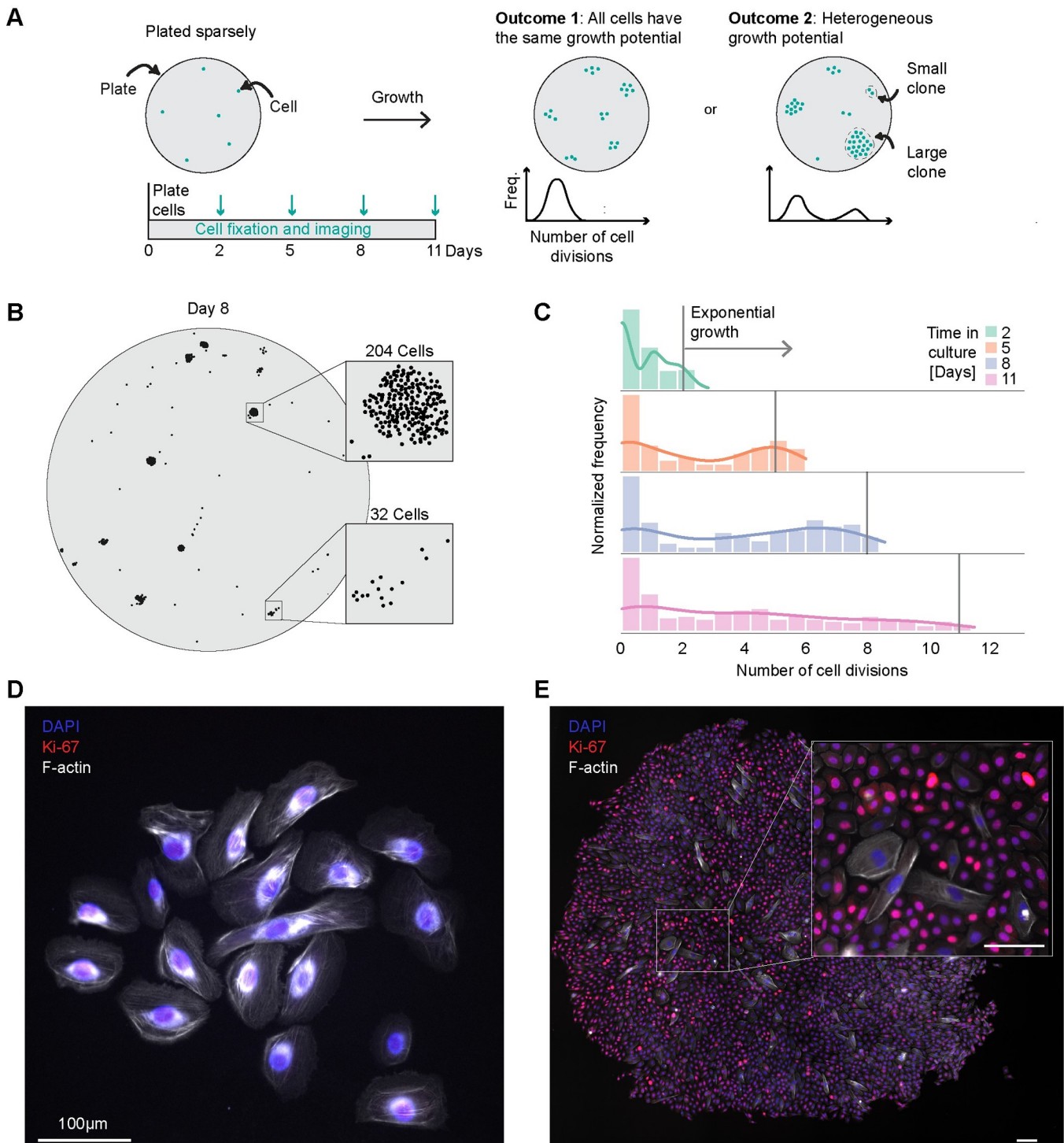

**Fig 1. Esophageal epithelial cells show different capacities for growth at low density.** (**A**) Schematic of experimental design. On the left, cells are plated at a low density, grown for 2, 5, 8 or 11 days, and imaged. (**B**) A representative plate of clones on day 8 of growth. Large and small clones are highlighted on the right with 249 cells and 20 cells, respectively. (**C**) Data from experiment in A. Histogram of the number of cell divisions, $\log_2$(number of cells per clone), separated by days in culture, as shown by the colors in the legend. The gray vertical lines show the cell number if they doubled every 24 hours (n = 2 biological replicates each with six technical replicates, one biological replicate shown with all technical replicates). (**D,E**) Immunofluorescence staining for Ki-67 and phalloidin staining for F-actin on day 11 with a small clone on the left and large clone on the right. Scale bars, 100 μm.

## Differences in growth dynamics between clones can be described by differences in a transition rate to a non-proliferative state

We next wanted to better understand the differences in growth dynamics for the highly proliferative and less proliferative clones. We developed a two parameter computational model for growth consisting of a doubling rate for the cells and a transition rate that describes the transition to a non-proliferative state (Fig 2A). We applied Gillespie's algorithm to simulate the growth of the highly and less proliferative clones (Fig 2B). To assess the model's accuracy, we analyzed its fit to the data from each clone type at day 8 (S3A Fig), using a range of parameters

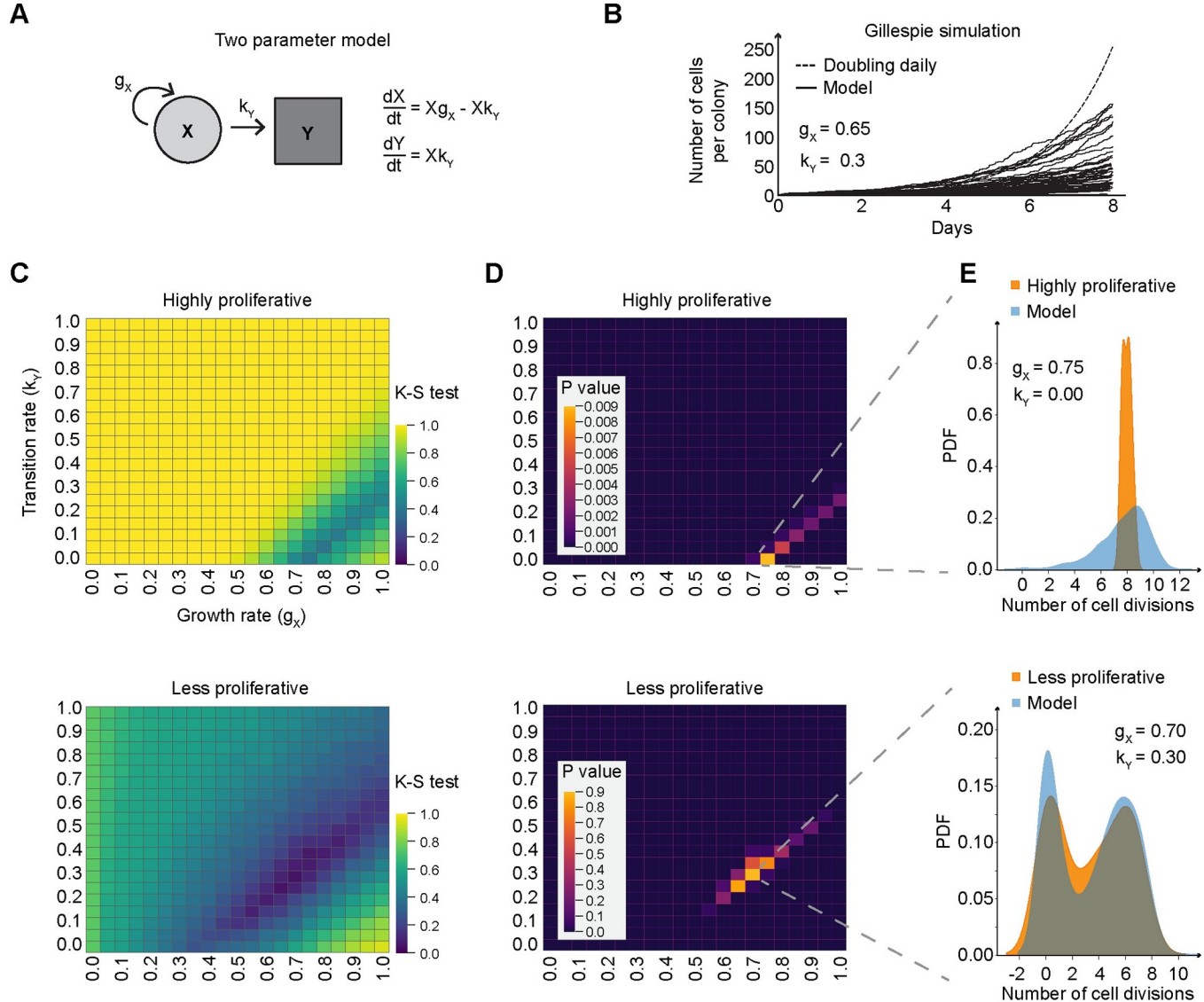

**Fig 2. Analysis of epithelial cell growth via gillespie simulations.** (A) Schematic of a two-state model, with a specified growth rate ($g_X$) and a transition rate ($k_Y$) and their respective ordinary differential equations. (B) Line plot of the outcomes of multiple Gillespie simulation runs based on the two-state model from panel A, with the growth parameter ($g_X$) set at 0.65 and the transition rate ($k_Y$) at 0.3. (C) Heat map of the simulated and experimental growth distributions at day 8, using the Kolmogorov-Smirnov test for each value in the parameter sweep. The top panel represents highly proliferative clones, while the bottom panel shows less proliferative clones. (D) Heat map displaying the p-values derived from the Kolmogorov-Smirnov test in panel A. (E) Probability Density Function (PDF) of log2(cell count per clone), separated by both experimental data (orange) and model predictions based on the optimal parameters from Panel B (blue). Two parameter sets are shown: g = 0.75 with k1 = 0 for highly proliferative clones and g = 0.7 with k1 = 0.3 for less proliferative clones.

(Fig 2C and 2D). The best model fit was determined by the lowest Kolmogorov-Smirnov (KS) test statistic and highest p-value, indicating the highest similarity between the modeled and observed distributions. The best fit model for the highly proliferative clones showed a growth rate of 0.74 and a transition rate of 0.0. In contrast, the best fit model for the less proliferative clones had nearly the same growth rate, but a significantly higher transition rate of 0.35–0.5 (Fig 2E). By applying this approach to all the time points in our data, the best fit model for the highly proliferative clones consistently had similar parameters, suggesting that these clones grow exponentially with a doubling time of 1.732 days (S3B Fig). However, for the less proliferative clones, the best fit model consisted of different parameters at different time points in the data (S3B Fig). This suggests that our two parameter model could not capture the growth dynamics of the less proliferative clones through time without drastically changing its parameters.

Building on our finding that the growth parameters of the less proliferative clones change through time, we turned to a logistic growth rate model to integrate time dependence. This model mirrors the observed pattern in our data: an initial surge in proliferative activity followed by a plateau as the cell population nears its carrying capacity (S3C and S3D Fig). For the less proliferative clones, this model achieved an optimal fit with a growth rate of 0.746, transition rate of 0.156 and carrying capacity of 126. With each of these models, we have assumed that cells uniformly progress through the cell cycle, that cells cannot re-enter the proliferative state once they have entered the non-proliferative state, and that death is negligible. From these models, we conclude that the growth dynamics of these two clone types are distinct and likely governed by different propensities for transitioning to a non-proliferative state.

## Clonal growth dynamics are largely heritable through four cell doublings

Considering that genetic factors do not explain the observed proliferation differences, we next investigated whether this proliferative phenotype is heritable through cell division. We hypothesized that while proliferative capacity was reversible on a 14-day timeline (S1C and S1D Fig), it might exhibit stability on shorter timescales. To test for heritability in the growth phenotype, we used DNA barcoding to track related cells and then measure their growth in our clonal assay. We transduced the EPC-hTERT cells with a high-complexity library of DNA barcodes that are integrated into the genome of single cells [38,39]. We used a low multiplicity of infection to ensure that each cell was labeled by only one barcode and then allowed the cells to proliferate through four doublings to generate roughly 16 copies of each barcode in the population. We next replated these barcoded cells randomly into two separate samples at low density. Our choice of four doublings ensured that each barcode would be represented in the two separate samples even with counting noise. We then allowed the sparsely plated barcode clones to grow over eight days, generating clones of different sizes as we previously observed. If the proliferative phenotype is not heritable, each clone should grow to different sizes in the two wells, indicating that growth is independent of the clonal identity (Fig 3A, Outcome 1). Conversely, if the proliferative potential of a clone is heritable, each clone should grow to a similar size in the two wells (Fig 3A, Outcome 2).

After eight days, we isolated genomic DNA from the samples and performed a targeted amplification of the barcode sequences. We quantified the number of reads for each barcode to use as a measurement of the clone size in each sample. To validate this method, we introduced defined numbers of cells containing known barcode sequences (50, 500, and 1000 cells) to each sample, including two replicates of the same cell number, but with different barcode sequences. As expected, the replicates were highly correlated (Pearson correlation 0.998) and corresponded to the expected numbers of cells in the assay (S4D–S4F Fig), verifying that read counts can reliably estimate cell quantities.

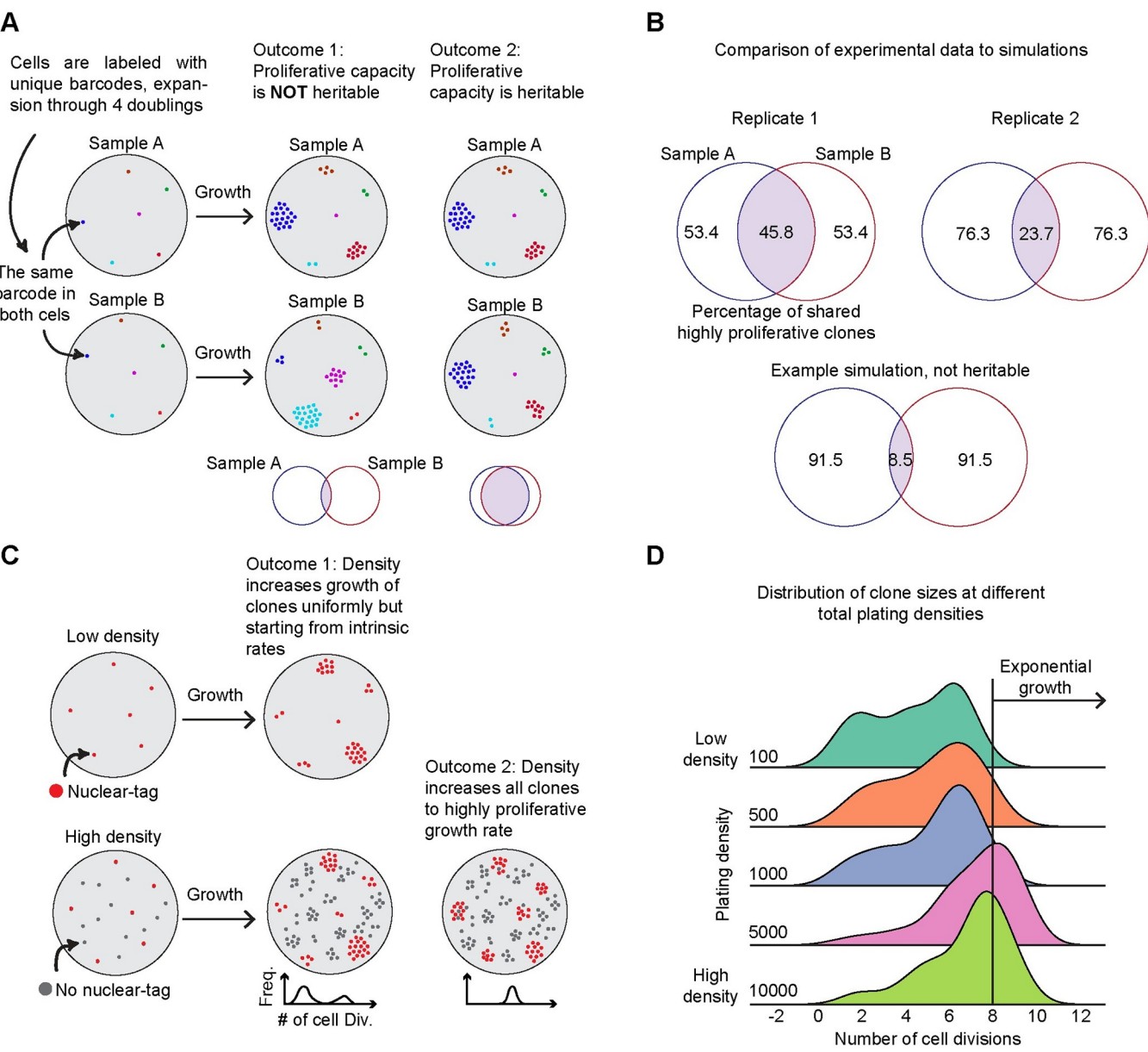

**Fig 3. Proliferative capacity is heritable and can be altered through extrinsic factors.** (**A**) Schematic of experimental design showing how lineage barcodes were used to extract information about clone growth. Potential outcomes for heritable and non-heritable states are shown on the right. (**B**) Venn diagrams comparing the overlap of the top 10% largest clones between sample A and sample B for two experimental replicates and a simulated dataset of random growth rates. (**C**) Schematic of experimental design showing mCherry-tagged cells being platted at a constant 50 cells per plate while non-fluorescent cells are seeded at 100, 500, 1000, 5000, or 10000 cells per plate. Outcomes from distinct densities and their effect on proliferation are shown on the right. (**D**) Histograms of cell divisions separated by their original plating density at day 8. Exponential growth is highlighted with a solid gray line (n = 2 biological replicates each with six technical replicates, one biological replicate shown with all technical replicates).

As in prior experiments, we took the top 10% of barcodes as the highly proliferative clones in each sample. We then asked whether the same highly proliferative clones exist in each of the samples. Across two replicates, we found that 45.8% and 23.7% of the highly proliferative clones were captured in each sample ([Fig 3B]). To test whether this overlap could be observed by chance, we simulated this experiment by generating an initial cell barcode population. We estimated the growth for each simulation by randomly choosing a growth rate from our

imaging data (Fig 1C). The simulated barcodes were then randomly divided into two groups and allowed to undergo five more divisions to model their growth as clones. We used the top 10% as the simulated highly proliferative in each sample and quantified the expected overlap in these barcodes. We found that the experimental overlap is consistently higher than the non-heritable simulations (Fig 3B).

We also analyzed the overall correlation in barcode abundance across the two samples (S4A and S4B Fig). The Pearson correlation coefficients were 0.511 and 0.373 for the replicates. In both analyses, deviations from a perfect correlation or perfect overlap could reflect a limited heritability of the growth phenotype as well as experimental factors such as differences in plating. In contrast, the Pearson correlation of our simulated random barcodes was 0.012, further confirming that the growth of these clones has a heritable component (S4C Fig).

## Extrinsic cues from neighboring cells can modulate the growth of clones

After establishing that heritable cell-intrinsic factors can influence a clone's growth potential, we next considered whether extrinsic factors could also have an effect. Numerous studies have shown that higher cell densities can enhance the growth of primary cells [40–42] and human cell lines [43,44]. This led us to investigate how density could alter clone size. However, this posed an experimental challenge because at higher cell densities, it is difficult to accurately track the growth of individual clones. To overcome this challenge, we generated wells with a consistent number of mCherry-labeled cells and varied the number of unlabeled cells. In our lowest density condition, we included 50 mCherry-labeled cells and 50 unlabeled cells. In our higher density conditions, we maintained the same number of mCherry-labeled cells (S4J FIg), but varied the unlabeled cells to reach a total of 100, 500, 1000, or 5000 cells per well. This setup allowed us to monitor the growth of the labeled clones, which could be easily identified due to their proximity on the plate. At each density, we allowed the cultures to grow for eight days and then measured the growth of each mCherry-tagged clone by imaging.

If increased cell density uniformly accelerated all clones' growth, we would expect differences in clone size to persist as they have varied intrinsic growth rates (Fig 3C, Outcome 1). However, if increased density caused all clones to match the growth of the highly proliferative, we would observe a more uniform distribution of clone sizes (Fig 3C, Outcome 2). Consistent with the previous experiments, at the lowest density, we found a range of different clone sizes, with 10% of clones following an exponential growth curve. As the overall density of the cultures increased, however, we found that the number of cells in each clone also increased (Figs 3D and S4I). We observed the largest average clone size at a plating density of 5,000 cells per well. At the highest density of 10,000 cells per well, we observed a slightly lower clone size, likely due to the culture approaching confluency. Using a complementary approach with DNA-barcoding, we further confirmed that higher cell densities generate more uniform growth among the clones (S4G and S4H Fig). Together, we concluded that the growth of individual clones can be modulated by increasing cell density, suggesting that external factors in the microenvironment can overcome the intrinsic differences between clones.

## Gene expression profiles of the highly proliferative and less proliferative clones are distinct

We next investigated gene expression between the less proliferative and highly proliferative clones. To simultaneously measure clone size and gene expression of single cells, we paired DNA barcodes with scRNA-seq (Fig 4A) [27,38]. We transduced EPC2-hTERT cells with this barcode library, plated them at a low density to capture the non-uniform clonal growth, and allowed them to grow for eight days. Afterward, we collected the cells for scRNA-seq and

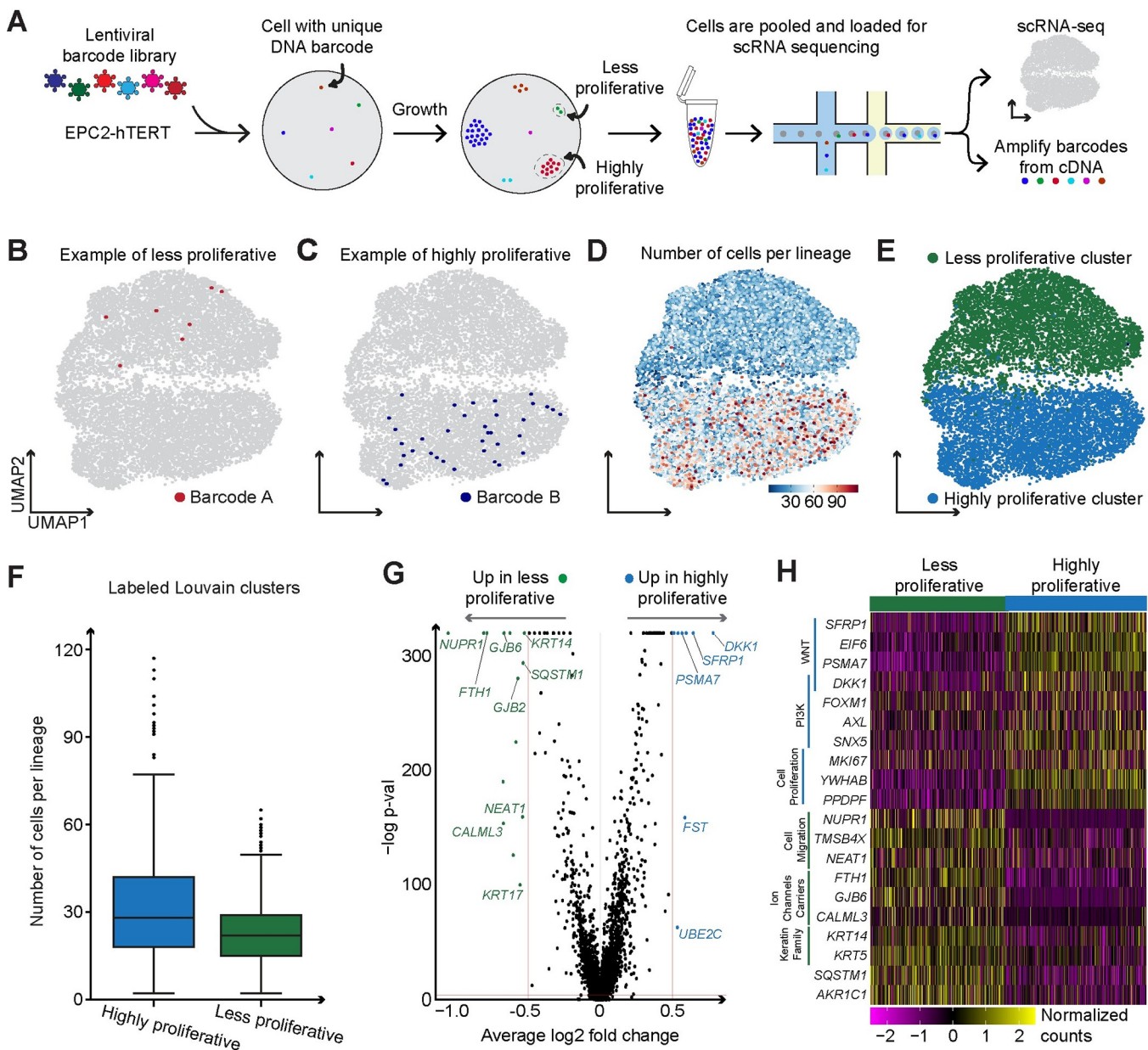

**Fig 4. Highly proliferative clones have a unique gene expression in which WNT and PI3K pathway genes are upregulated.** (**A**) Schematic of how lineage barcodes were used in combination with scRNA-seq. (**B,C**) UMAP plots with an example lineage from less proliferative and highly proliferative clones respectively. (**D**) UMAP plot overlaid with the number of cells per lineage barcode. (**E**) UMAP plot of all lineage barcodes labeled as less proliferative or highly proliferative. (**F**) Box plots of the number of cells per lineage in each clone type. (**G**) Volcano plot of differential expression analysis between highly proliferative (positive fold change) and less proliferative (negative fold change). (**H**) Heatmap of the log10 normalized and scaled gene expression of highly proliferative and less proliferative cells.

recovered the transcribed barcodes from the cDNA libraries. We matched the barcodes to the transcriptomes of single cells, identifying lineages of different sizes. We then visualized this data using uniform manifold approximation and projection (UMAP) (Figs 4B, 4C, S5A, and S5B). We labeled cells based on their clone sizes, using a 10% threshold determined by our previous observations S1B Fig. Cells belonging to the largest 10% of lineages were categorized as highly proliferative clones, whereas those in smaller lineages, comprising the remaining 90%,

were categorized as less proliferative lineages. After identifying these highly proliferative and less proliferative populations, we found that they were separated consistently from each other across multiple dimensionality reduction methods, indicating a robust distinction in high-dimensional space (S5C and S5D Fig), and this clustering was not driven by batches within our 10x library samples (S6A–S6C Fig).

We also noticed that UMAP showed a strong separation of cells based on their predicted phase in the cell cycle (S5E Fig). However, we found that the cell cycle differences did not align with the differences between highly proliferative and less proliferative clones. Both populations of cells contained cells in each phase of the cell cycle, indicating that the differences in gene expression between these populations are not predominantly driven by cell cycle genes. This finding is consistent with our modeling results in which the less proliferative clones still retain some proliferative cells at day 8. To control for cell cycle effects in our gene expression analysis, we regressed out the cell cycle using previously identified S-phase and G2/M-phase gene sets [45], ensuring that the observed gene expression changes reflected biological differences beyond cell cycle variations. Next, we overlaid the number of cells per lineage barcode onto the UMAP (Fig 4B–4D). This representation allowed us to clearly visualize and label clusters that belonged to either the highly proliferative or less proliferative state (Fig 4E). These newly labeled clusters continued to exhibit the expected trend with larger clones residing in the highly proliferative cluster even after removal of cell cycle effects (Fig 4F).

Upon comparing the gene expression of cells in the highly proliferative and less proliferative clones, we found significant differences in gene expression (Fig 4G and S2 Table), with cells from highly proliferative clones having a higher expression of *MKI67*, as expected. We also found that cells from highly proliferative clones had higher expression of multiple WNT pathway genes, including *SFRP1*, *DKK1*, *EIF6*, and *PSMA7*, and PI3K pathway genes, including *FOXM1*, *AXL*, and *SNX5* (Fig 4H). Intriguingly, a previous study showed that *DKK1* and *FOXM1* can form a positive feedback loop that promotes cell growth in esophageal squamous cell carcinoma and pancreatic cancer [46]. We found this same co-expression pattern in cells from the highly proliferative clones, which might suggest that a similar feedback loop could be maintaining the proliferative state in these cells.

While scRNA-seq showed expression of PI3K pathway genes in the highly proliferative clones, we next wanted to test whether inhibiting PI3K would affect the growth of these clones. We sparsely plated the EPC2-hTERT cells as in other experiments and treated them with PI3K inhibitor, GDC-0941, for 8 days. This treatment led to a noticeable decrease in the number of cells in each clone, supporting that PI3K activity is needed for growth in these clones (S7A Fig). However, since both clone types are inhibited by the PI3K inhibitor in this experiment, the differential PI3K activity between the highly and less proliferative remains to be experimentally validated.

Our scRNA-seq data also pointed to DKK1 as upregulated in the highly proliferative clones. To functionally test whether secreted DKK1 affects growth, we treated cells with varying concentrations of recombinant human DKK1. We found no significant differences across the tested concentrations (S7B Fig). To further explore the potential role of DKK1 in clonal growth, we created a DKK1 knock-out (KO) cell line and found that this knockout increased the percentage of highly proliferative clones when compared to the *Rosa26* control (S7C–S7E Fig), although it did not make every clone grow exponentially. We thus reasoned that DKK1 might shift the balance in the states regulating the highly proliferative and less proliferative clones. Together, we found that both inhibiting PI3K activity and knocking out DKK1 can modulate the proliferative phenotypes that underlie these clonal growth behaviors.

### The less proliferative clones express KRT14, a marker of quiescent basal cells in the esophagus

We next examined the transcriptional signatures of the less proliferative clones and found that they expressed higher levels of *KRT14, DST and COL17A1* (Fig 5A), markers indicative of quiescence in the human esophagus [47–50]. We thus investigated whether KRT14 was marking the non-proliferative subset of cells in the less proliferative clones, specifically those that appear flat and elongated. We grew clones at low density for 8 days in 6-well plates, and performed immunofluorescence to stain for KRT14 and Ki-67. The highly proliferative clones showed a significant presence of Ki-67 and a minimal expression of KRT14 (see Fig 5B, top). Meanwhile, the less proliferative clones showed the opposite trend—low Ki-67 expression accompanied by a relative increase in KRT14 levels (Fig 5B, bottom). We compared the distribution of these markers in cells from the largest clone with cells from the three smallest clones in this

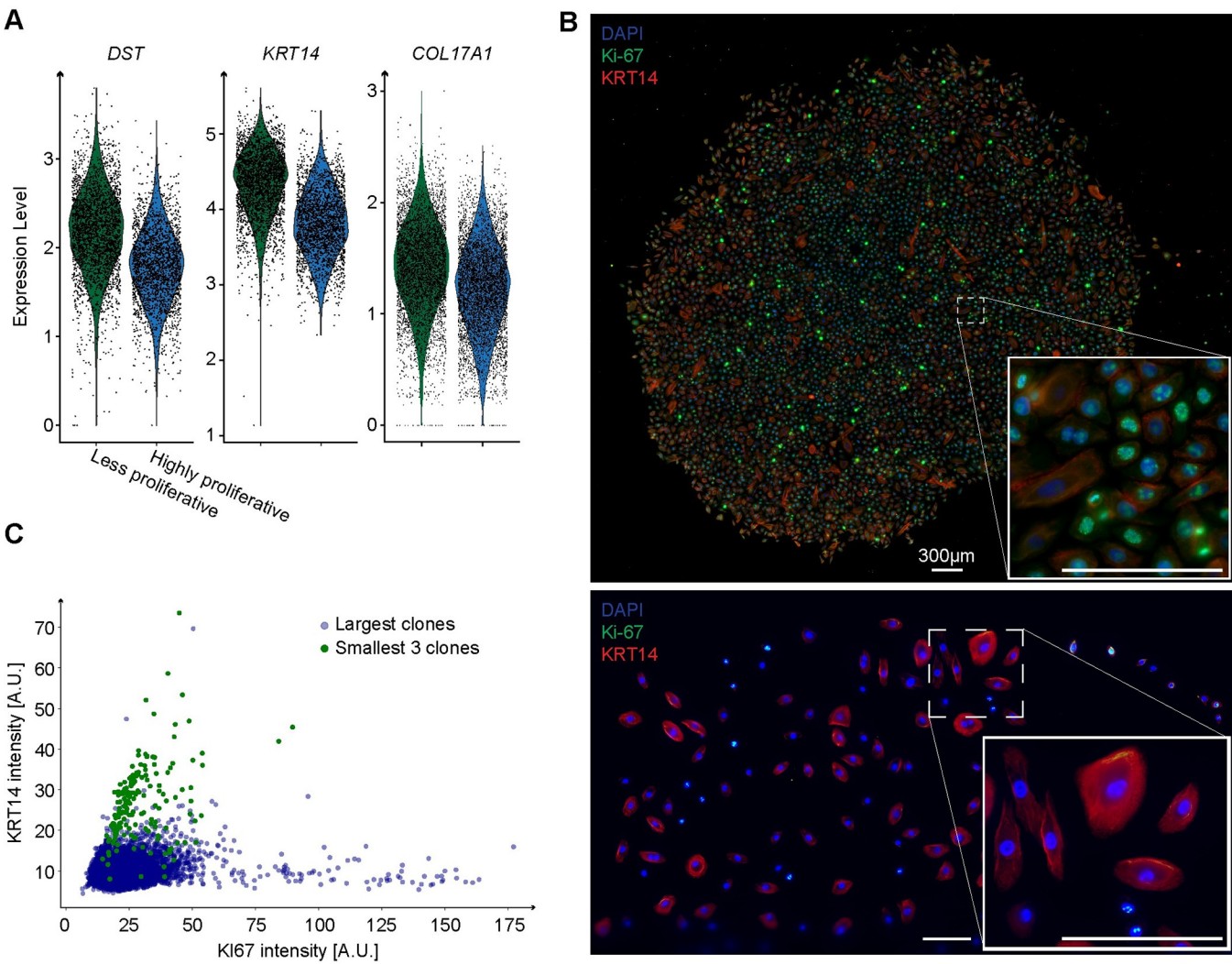

**Fig 5. KRT14 expression is enriched in less proliferative clones and marks large flat cells.** (**A**) Violin plot showcasing the distribution of expression levels for DST, KRT14 and COL17A1 which are basal quiescent markers of the esophageal epithelia. The data is derived from EPC2-hTERT scRNA-seq. (**B**) Representative image captured on day 11 showing a large clone at the top and small clone at the bottom, stained for Ki-67 (proliferation marker) and KRT14 (keratin 14, an epithelial quiescent marker). All scale bars represent 300μm. (**C**) Scatter plot of Ki-67 and KRT14 intensities on the x-axis and y-axis, respectively, comparing the largest clone with three of the smallest clones (n = 2 technical replicates).

experiment (Fig 5C). Consistent with our EPC2-hTERT scRNA-seq data, we found much higher KRT14 expression in cells derived from the smallest clones (S6D and S6E Fig). Furthermore, we noticed that the cells which appeared elongated and flattened consistently demonstrated a higher abundance of KRT14. The presence of high KRT14 in some cells of the highly proliferative clone, though less frequent, could suggest that these cells have transitioned towards a less proliferative state. These findings support our model in which the less proliferative clones consist of a quiescent subset of cells that are marked by KRT14 expression and an elongated morphology.

## A rare subset of cells with a highly proliferative transcriptional signature was found in the human esophageal epithelium

We next wondered whether the gene expression signatures derived from the EPC2-hTERT cell line model are enriched in a particular compartment of cells within the healthy human esophagus. We collated and analyzed four published scRNA-seq datasets of healthy human esophageal epithelium [51–54]. After quality control filtering, we independently clustered and labeled each dataset based on known cell type markers: basal (*DST*, *KRT15*), suprabasal dividing (*MKI67*, *TOP2A*), suprabasal non-dividing (*DSC2*, *DSP*), intermediate (*SERPINB3*, *PADI1*, *TGM3*), and superficial (*MT1G*, *RNASE7*, *KRT16*) (Fig 6A) [51–53]. We then merged the labeled datasets of each sample, 62,216 cells across 24 patients, and batch-corrected the combined data (S8A and S8B Fig).

To test whether these compartments express the genes in our highly proliferative clones, we created a highly proliferative cell score based on nine genes that were selected as differentially expressed between the highly and less proliferative clones using a cut-off of log2 fold change greater than 0.5 and adjusted p-value of less than 0.05. We then calculated the activity for this gene set for every cell in the combined scRNA-seq dataset (Fig 6B) and split the cells by their predicted esophageal epithelial layer. We found that the highly proliferative signature is most

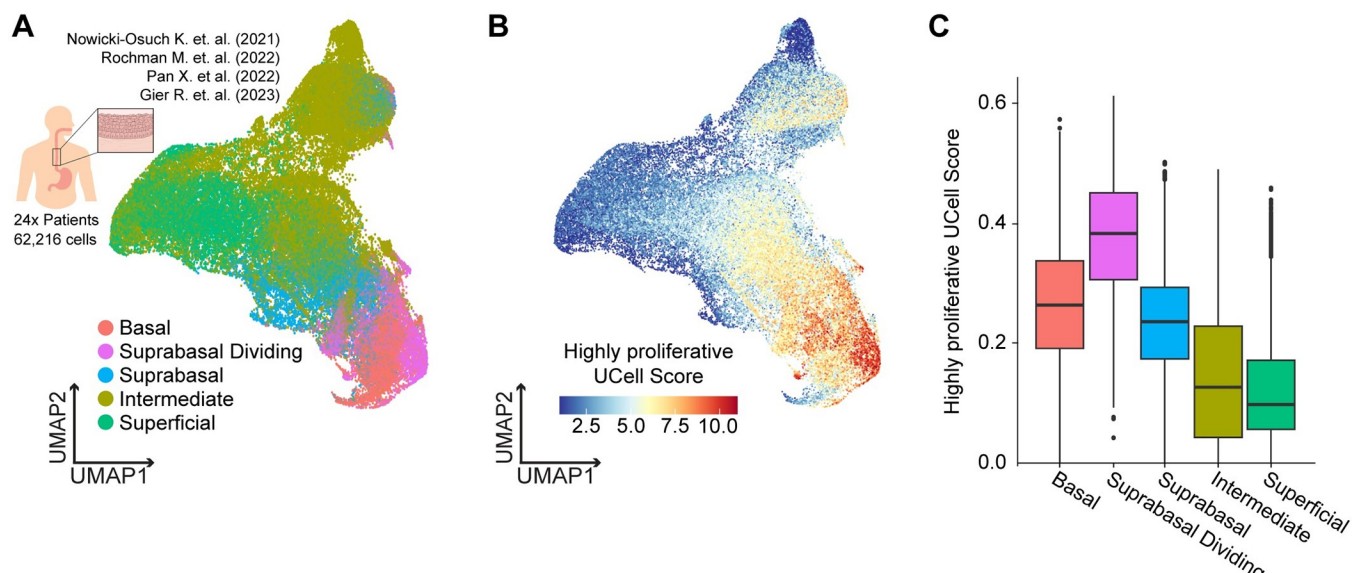

**Fig 6. The highly proliferative clone signature shows the highest enriched in the suprabasal dividing compartment of the human esophagus.** (**A**) UMAP plot of merged 24 patients and 62,216 cells clustered by tissue-specific cell types. (**B**) UMAP plot illustrating merged patient scRNA-seq data, with each cell color-coded based on the highly proliferative gene set signature derived from EPC2-hTERT cells. (**C**) Box plot of the highly proliferative signature grouped by different tissue-specific cell types.

strongly expressed in the suprabasal proliferative cells (Fig 6C). In the human esophagus, these cells sit directly above the basal cells and consist of the most proliferative subset of the epithelial tissue (S8C–S8D Fig) [9,48]. Importantly, this signature was derived after cell cycle regression and does not contain markers of proliferation. Therefore, the enrichment for the highly proliferative clone signature in the suprabasal proliferative compartment highlights a potentially shared gene expression state between this in vitro model and the human esophagus.

## Discussion

In this study, we find that single epithelial cell clones exhibit a wide range of growth potentials. Many clones either slow or halt their growth, but a notable subset grows substantially. Our mathematical models revealed that these growth dynamics can be characterized by exponential growth in the highly proliferative clones and the inclusion of a logistic growth rate for the less proliferative clones. Additionally, we found that this growth potential is heritable through cell division, yet can be modulated through altering cell density. Through analyzing gene expression differences in clones of different sizes, we identified the specific cellular states that are differentially activated in the highly proliferative clones. Ultimately, we identified a similar highly proliferative cellular state enriched within the proliferative suprabasal layer in the human esophagus scRNA-seq.

Although our study's immortalized human esophageal cell line may not fully encapsulate the complexity seen in vivo, it may still capture key aspects of the human esophageal epithelium. In the human esophagus, progenitor cells proliferate to generate differentiated cells that sit above them within the tissue. Thus, it is conceivable that characteristics of our non-proliferative cells might be similar to in vivo differentiation. Similarly, other esophageal modeling studies have incorporated a differentiated cell state that is no longer able to grow [18,55–57]. While we do not explicitly assume that non-proliferation reflects differentiation, our central conclusion is the same that the less proliferative clones have the ability to generate cells that no longer grow, while the highly proliferative clones do not. Furthermore, the variability that we observe between clones could also be related to the differences seen in progenitor populations in the human esophagus. Specifically, the human esophageal epithelium contains two distinct progenitor populations. A slow cycling progenitor that plays a crucial role in responding to environmental insults [8,58] and maintaining genome stability [59,60]. The other progenitor driving proliferation, contributing to tissue growth and renewal [61–63].

This study adds to the expanding body of work on non-genetic cellular variability and cell fate. Here, we focus on the proliferative capacity of each clone, measured by imaging individual clone sizes after a growth period. We find that this growth fate is not encoded by genetic differences between cells and that both intrinsic and extrinsic factors can contribute to clonal growth fates. Using DNA barcoding technologies, we show that the cell states that mediate these growth behaviors are partially heritable through a limited number of cell divisions, similar to observations of memory in cancer drug resistance [27,30,38], immune cells [64], and differentiation [30,65,66]. While we describe the cell states associated with these phenotypic differences in epithelial progenitor cells, the mechanisms that encode the cellular state memory remain to be uncovered.

## Methods

### Cell lines and culture

Human esophageal epithelial cells (EPC2-hTERT) were a gift from A. Muir (Department of Pediatrics at Children's Hospital of Philadelphia, USA). Integration of the hTERT gene into EPC2 cells is presumed to be heterogeneous in the cell line. Nonetheless, our experiments have

consistently yielded a uniform distribution of clone sizes, this uniformity persists even in cultures initiated from a single clone. STR profiling was performed by the Penn Genomic Analysis Core. We cultured EPC2-hTERT in Keratinocyte SFM (KSFM) supplemented with 2.5 μg of human recombinant epidermal growth factor (rEGF), 25 mg of bovine pituitary extract (BPE) (Gibco, 17005042), 50 Units/mL penicillin, and 50μg/mL streptomycin. Cells were passaged at 80% confluency using TrypLE Express Enzyme (Gibco, 12604013) and neutralized with equal amounts of 0.5mg/mL soybean trypsin inhibitor (STI; Gibco, 17075029). Human embryonic kidney cells (HEK293T) were cultured in 10% DMEM (90% DMEM high glucose with GlutaMAX (Gibco, 10569010), 10% FBS (GE, SH30396.03), 50 Units/mL penicillin, and 50μg/mL streptomycin (Gibco, 15070063). They were passaged with 0.05% trypsin-EDTA (Gibco, 25300054). All cells were incubated at 37˚C and 5% $CO_2$. For all experiments reported in this manuscript, cell cultures were consistently used between passages 40 and 48 to ensure uniformity in cell behavior. A new vial of the cell line was thawed whenever the culture neared passage 48, maintaining consistency across experiments. Biological replicates were performed with different cultures of cells and on different dates. Technical replicates were derived from the same culture and on the same date.

## Single-cell clone isolation and subcloning

EPC2-hTERT cells were thawed and passaged twice before each experiment. To isolate single cells, we counted cells and diluted to a concentration yielding 0.5 cells per well when plated into a 96-well plate. After 8 hours of incubation, each well was examined under a microscope. Wells containing exactly one adherent cell were marked and tracked; wells with zero or multiple cells were excluded from further analysis. Cells were cultured for 14 days total with the media replaced every four days. After 14 days, the single-cell derived clones were classified based on their growth: highly proliferative clones were those that reached >90% confluency, while less proliferative clones were less confluent.

We subcloned a subset of the highly proliferative and less proliferative clones. For subcloning, we detached all cells in a single-well using TrypLE as described in "Cell lines and culture section". We then diluted and replated these cells into a 96-well plate as single cells. This workflow allowed us to take all the cells from one clone and then redistribute them as single cells again. We performed this subcloning for seven highly proliferative clones (from 3 biological replicates with 3, 2, and 2 technical replicates in each) and three less proliferative clones (from 3 technical replicates).

## Multi-day clonal fixation experiment

We plated cells in six-well plates with 50 cells per well. Cells were grown for 2, 5, 8 or 11 days (Figs 1C and S3A) or 2, 4, 6, 8, 10 or 12 days (2 biological replicates with 6 technical replicates per day). Media was changed every 4 days. After the indicated time had passed, we fixed the cells by aspirating the media and washing twice with 1X Dulbecco's phosphate buffered saline (DPBS; Corning, 21-031-CV), followed by a 10-minute treatment with 3.7% formaldehyde (Fisher, BP531-500) in PBS (Invitrogen, AM9625). The formaldehyde was aspirated, and the wells were washed twice with DPBS. We then added 2 mL of 2x saline-sodium citrate (SSC) to each well and stained with DAPI. Imaging of the wells was then performed using a 10× objective on a Nikon Eclipse Ti2 fluorescence microscope. Clones with one cell were filtered out of the final analysis.

## Cell and clonal counting

To evaluate cell counts post-fixation, we utilized Cellori, a nuclei counter which can be accessed at https://github.com/zjniu/Cellori/tree/main and DeepCellHelper (https://github.

com/SydShafferLab/DeepCellHelper). To identify individual clones from the output of Cellori and DeepCellHelper, we developed a custom graphical user interface, ColonySelector, which can be found at https://github.com/SydShafferLab/ColonySelector. By circling individual clones in each well using ColonySelector, we generated a comprehensive record associating each nucleus with its respective clone. Through the integration of these methodologies, we achieved a precise and efficient analysis of clonal counts. Clones varied significantly in size, therefore we implemented a $\log_2$ transformation (referenced as number of cell divisions) to allow for a more effective comparison and visualization of the number of cells per clone that contained vastly different sizes.

## Bimodality coefficient analysis

We utilized the GaussianMixture class from the scikit-learn Python package to fit a GMM to the clonal growth rate data. The model was trained using the expectation-maximization algorithm to identify distinct Gaussian distributions within the dataset. The optimal number of components was selected based on the Bayesian Information Criterion (BIC). We applied this algorithm to all time points for each biological replicate (S2 Fig). To quantify bimodality, we calculated the bimodality coefficient from the skewness and kurtosis of the data. Bimodality coefficients above 5/9 suggest bimodality, below 1/3 suggest unimodality, and values between are considered ambiguous. Each Gaussian component was assigned a weight, representing the proportion of the overall data distribution it explains, ensuring that the sum of the weights equals one.

## Quantitative modeling of epithelial cell proliferation dynamics

We employed mathematical modeling in python using the SciPy library to accurately represent the growth dynamics of epithelial cells. To account for the stochastic nature inherent in biological processes, we used the random function from the random library to implement the Gillespie simulation to model epithelial cell growth. This was accomplished by using a stoichiometry matrix for cell proliferation and transition to a non-proliferative state (Fig 2B). The simulation involved calculating reaction propensities and updating the system state based on these reactions, constrained by a maximum time length. This process continued until either no more reactions could occur (all cells are in a non-proliferative state) or the maximum time length was reached. This method allowed us to capture the stochastic nature of biological processes and compare simulation results with experimental data using the Kolmogorov-Smirnov test, scipy.stats.ks_2samp, to assess model fit in Fig 2C and 2D. Consistently, the results of these tests indicated that a more complex model might be required to fully capture the observed growth behavior of the epithelial cells.

We developed an exponential growth model and a logistic growth rate model described by ordinary differential equations (ODEs), seen in S3C Fig. We then used Scipy.integrate.odeint to solve the system of ODEs for each. We used a cost function that integrates this ODE system using odeint and calculates the sum of squared differences between the model's output and experimental data, acting as a metric for the goodness of fit. Parameter optimization is achieved through the scipy.optimize.minimize function, which seeks the best-fit parameters ($g_X$ and $k_Y$) that minimize the cost function. Upon determining the optimal parameters, we used odeint to solve the ODE system with these optimized parameters, generating the best-fit solution of the model, S3D Fig.

## Live cell imaging and analysis

We used the IncuCyte S3 system (Sartorius) for our time-lapse experiments. The cells were diluted to 0.5 cells per well in a 96-well plate, and allowed at least 12 hours to adhere to the

plate before imaging. We used either wild type EPC2-hTERT cells or a genetically modified EPC2-hTERT to express H2B-mCherry (see Lentiviral transduction section), to facilitate nuclear tracking. We captured images at 4x magnification with a 300 ms exposure time for brightfield every 12 hours and 300 ms exposure time for mCherry every 24 hours, respectively (2 technical replicates for each). The nuclear masks were generated using IncuCyte's built-in software, from which we exported CSV tables detailing either the mask of cell area (brightfield) or the count of nuclei in each well (mCherry) at various time points. The clones in each mask were then labeled using the ColonySelector (https://github.com/SydShafferLab/Colony Selector). This data was subsequently plotted using python programming.

### Immunofluorescence and phalloidin staining

EPC2-hTERT were washed twice with 1X Dulbecco's phosphate buffered saline (DPBS; Corning, 21-031-CV) and fixed with 3.7% formaldehyde (Fisher, BP531-500) in PBS (Invitrogen, AM9625) for 10 min at room temperature. We then washed twice with PBS and permeabilized with 70% ethanol for at least 1 hour. Cells were blocked with 1% BSA in PBS for 1 hour at room temperature. Primary Ki-67 (SP6) rabbit monoclonal antibody (Sigma-Aldrich, SAB5500134) and primary KRT14 (LL002) mouse monoclonal antibody (Abcam, ab7800) were diluted in 0.1% BSA in PBS (1:200 dilution). Primary staining solution was added and incubated overnight at 4°C. Secondary Alexa Fluor 647 goat anti rabbit IgG H+L (A21244) and Alexa Fluor 488 Donkey Anti-Mouse secondary antibody (Jackson Labs, 715-545-151) were diluted in 1% BSA in PBS (1:500 dilution) and, when required, combined with 7.5 μL stock solution of Phalloidin-Atto 488 (Sigma-Aldrich, 49409) for a total of 1.5mL 1% BSA in PBS. After two PBS washes, the secondary staining solution was added for 1 hour at room temperature in the dark. After two more PBS washes, we added 2x saline-sodium citrate (SSC) with DAPI and either imaged immediately or stored at 4°C prior to imaging.

### Lentivirus production

To transfect HEK293T cells with the lentiviral plasmid, we first grew them to 90% confluence in a 10cm dish. We then prepared two tubes of transfection reagents: one containing 500μl of OPTI-MEM and 80 μl of 1 mg/mL PEI, and another containing 500μL of OPTI-MEM, 9 μg of psPAX2 plasmid, 5.5 μg of VSVG plasmid, and 8 μg of the barcode plasmid. After combining the contents of the two tubes, we incubated the mixture at room temperature for 20 minutes before adding it dropwise into the HEK293T plate. Following incubation of the cells at 37°C for 7 hours, we removed the medium and washed the plate once with DPBS. Next, we applied 6 mL of fresh medium and incubated the cells at 37°C for approximately 12 hours before repeating the medium collection process every 12 hours for a total of 72 hours. We filtered all the collected medium through a 0.2 μm filter and stored it at -80°C.

### Lentiviral transduction

Cells were transduced by creating a mixture consisting of polybrene (final concentration of 4 μg/mL), virus (determined by titration to be 20% infection), and cells at a concentration of 200,000 cells/mL. We then added 2 mL of this mixture to each well of a 6-well plate and subjected the plate to centrifugation at 600 RCF for 25 minutes. We incubated the cells with the virus at 37°C for 12 hours, after which we removed the virus-containing media and washed with DBPS. Subsequently, we added 2 mL of fresh medium to each well. The following day, we transferred each well to a 10 cm dish and allowed the cells to express the construct for 2–3 days.

### Fluorescence-Activated Cell Sorting (FACS)

We used TrypLE to detach the cells from the plate and STI to inhibit the TrypLE. The dissociated single-cell suspension was then washed once with 0.1% BSA and resuspended with 1% BSA and kept on ice. All cells that were sorted had a fluorescent tag that was detectable by the Beckman Coulter Moflo Astrios. We used a 100 μm nozzle and forward and side scatter to gate for single cells. We then sorted out the cells expressing the fluorescent tag, and WT cells were used as a negative control to determine gatting.

### Heritability experiment

The EPC2-hTERT were transduced with the DNA barcode library at a low MOI, achieving single-barcode integration per cell. We then allowed the cells to proliferate over four doublings, yielding approximately 16 copies of each barcode in the population. Barcoded cells were then randomly replated and split into two separate wells at a low density to generate clones. Clones were grown for 8 days, changing the media every 4 days (2 technical replicates). Genomic DNA was then retrieved from both splits, following the protocol described in the section "scMemorySeq: lineage barcode recovery from scRNA-seq and gDNA".

### Heritability analysis

We simulated data and calculated the Pearson correlations for three distinct growth rate probability distributions: unimodal, bimodal, and stochastic; where each cell picked a random growth rate from the distribution. These simulations are based on the assumption that sibling cells select their growth rates randomly, implying that they do not retain any memory of their previous proliferation rates. For the unimodal growth rate distribution, we estimated the parameters of a Gaussian distribution (specifically the mean (μ) and standard deviation (σ)) that best approximate the overall distribution on the day 8 data (Fig 1C). We fit the bimodal growth rate distribution using scipy.stats.gaussian_kde from the python scikit-learn library. Finally, stochastic growth was assigned equal probability to all possible growth rates determined by experimental data of day 8 (Fig 1C). Each simulation was run 1000 times to generate the distribution of Pearson correlations seen in S4C Fig.

### Clonal density experiments

We explored the effects of density on clonal proliferative capacity through two orthogonal experiments: mCherry nuclear tag and barcode sequencing.

First, the EPC2-hTERT were transduced with H2B-mCherry nuclear tag to enable easier detection and quantification of cells. mCherry-tagged cells were plated at a constant 50 cells per plate while non-fluorescent cells were seeded at 100, 500, 1000, 5000, or 10000 cells per plate through dilutions. Cells were then grown for 8 days and media was changed every 4 days (2 biological replicates each with 6 technical replicates). Cells were then fixed with 3.7% formaldehyde for 10 min. Formaldehyde was aspirated, washed twice with DBPS and permeabilized with 70% ethanol for at least 1 hour. We then washed twice with DBPS and added 2mL of 2xSSC with DAPI to each well. Each well was then imaged using a 10× objective on a fluorescence microscope (Nikon, Eclipse Ti2).

Second, the EPC2-hTERT were transduced with the DNA barcode library as described above. Barcoded cells were sorted to reach either 500, 1000, 5000, or 10000 cells per plate. Cells were then grown for 8 days and media was changed every 4 days (2 technical replicates). At day 8 genomic DNA was retrieved from each seeding density, following the protocol described

in the section "scMemorySeq: lineage barcode recovery from scRNA-seq and gDNA". S4G and S4H Fig shows that the read depth of the barcodes is equal throughout distinct densities.

## scMemorySeq: Lineage barcode recovery from scRNA-seq and gDNA

Clonal dynamics were extracted using a high-complexity transcribed barcode library as previously described [27,38,67], with the library generation protocol available at: https://www. protocols.io/view/barcode-plasmid-library-cloning-4hggt3w. Briefly, a CRISPR-Cas9 guide RNA plasmid (LRG2.1T) was used as a backbone for the DNA barcode. The plasmid was altered by replacing the U6 promoter and the single guide RNA scaffold with an EFS promoter and GFP. This was preceded by 100 semi-random nucleotides consisting of WSN repeats (W = A or T, S = G or C, N = any) that created the barcodes. The final sequence for the DNA barcode plasmid can be found here: https://benchling.com/s/seq-DAMUWPyU198hRSbpiecf.

The lineage barcodes are recovered from the excess full-length cDNA aliquot produced from the 10x library protocol. We PCR amplified the lineage barcodes with primers that were compatible with the NextSeq 500 and that would amplify the lineage barcode and the 10x cell barcode. The amplification was done by combining 100 ng of full-length cDNA per reaction, 0.5μM of each primer, and PCR master mix (NEB, M0543S). The amplification began with a 30-second denature step at 98˚C, then 98˚C for 10 seconds followed by 65˚C for 2 minutes repeated 12 times, and a final extension step of 5 minutes at 65˚C. We then performed SPRI bead (Beckman Coulter, B23317) size selection (using a 0.6X bead concentration) to extract the amplified barcodes (~1.3kb). Finally, the lineage barcode library was sequenced with a NextSeq 500 with a Mid Output Kit v2.5 (150 cycles, Illumina, 20024904) by paired-end sequencing using 8 cycles on each index, 28 cycles on read 1 to read the 10x barcode and UMI, and 123 cycles on read 2 to sequence the lineage barcode.

The scRNA-seq and lineage barcode data was combined following the scMemorySeq custom pipeline that creates custom FASTQ files for the lineage barcodes that can be inputted in conjunction with the 10x Genomics Cell Ranger Feature Barcode pipeline. The custom python script is available through GitHub: https://github.com/SydShafferLab/BarcodeAnalysis.

Total genomic DNA was extracted from the samples using the QIAamp DNA Mini kit (Qiagen, 56304) following the manufacturer's instructions. The target barcode region was amplified using PCR with primers targeting the region of interest that contained the Illumina adapter sequence, and index sequences. PCR amplification was carried out in a reaction mixture containing PCR Master Mix (NEB, M0543S), 0.5 μM of each primer, and 500 ng of template DNA. The amplification protocol consisted of an initial denaturation step at 98˚C for 30 seconds, followed by 24 cycles of 95˚C for 10 seconds and 65˚C for 40 seconds, with a final extension at 65˚C for 5 minutes. The PCR products were size excluded with SPRI beads (Beckman Coulter, B23317) to prepare the DNA library for sequencing using a double-sided cleanup. We then sequenced on an Illumina NextSeq 500 with a Mid Output kit (150 cycles, Illumina, 20024904) for single-end sequencing (151 cycles for read 1, and 8 cycles for each index).

## Single-cell RNA sequencing EPC2-hTERT

We generated scRNA-seq libraries in accordance with the manufacturer's protocol (10x Genomics Chromium Next GEM Single Cell 3' v3.1 Dual Index kit). For the esophageal epithelial cells (data shown in Fig 3B and 3C), we sorted out 3,600 cells with lineage barcodes (GFP positive cells) and plated 50 cells per well in six 6-well plates and let them expand for 8 days, changing the media every 4 days. We then loaded the single-cell suspensions onto a 10x Chromium Controller to generate GEMs for each sample. We aimed to generate approximately 20,000 cells between two lanes while taking into account the estimated recovery of 50%. Finally, we

sequenced all our libraries with paired-end sequencing on an Illumina NextSeq 550 using 10 cycles for both indices, 28 cycles for Read 1, and 43 cycles for Read 2.

## Single-cell RNA data mapping, analysis and filtering

We generated the FASTQ files for the EPC2-hTERT dataset using CellRanger mkfastq v5.0.0, which processed raw Illumina base call files. The resulting FASTQ files were then aligned to the 10x reference GRCh38-2020-A using STARsolo v2.7.9a [68]. For downstream analyses, count matrices were generated utilizing the -soloFeatures GeneFull argument for each lane. Each dataset was individually filtered based on standard filtering before combining them using Seurat V4 [69]. First, we excluded poorly sequenced cells with a low number of genes detected. Additionally, we removed cells that were likely to be doublets, characterized by a high number of genes. We then removed cells based on their mitochondrial percentage to eliminate low-quality or dying cells from our dataset. We also used a doublet finder (scDblFinder) to further filter doublets [70]. Each individual dataset was then independently normalized with NormalizeData.

The EPC2-hTERT scRNA-seq dataset had no distinguishable bash effects (S5D and S5E Fig), we therefore proceeded with standard dimensionality reduction, FindVariableFeatures (based on the 2,000 most variable features), ScaleData,RunPCA, FindNeighbors (20 dimensions) and FindClusters. UMAPs were then generated using the top 20 principal components. DNA barcodes were then used to determine the highly proliferative clones in the scRNA-seq data. We used the top 10% largest clones as our threshold for highly proliferative clones. Cells containing no barcodes (7893 cells), as well as cells with four or more barcodes (228 cells) were discarded.

We then used DNA barcodes to label clusters after regressing out the cell cycle, "*Single-cell RNA sequencing cell cycle analysis*". This highlighted a separate population with a unique gene expression state that we labeled as the less proliferative clones. We then performed differential gene expression analysis between these newly labeled highly proliferative and less proliferative clusters (S2 Table). Finally, we selected the highly proliferative gene signature set to include all genes exhibiting a positive log2 fold change greater than 0.5 and adjusted p-value of less than 0.05 among the differentially expressed genes identified between highly proliferative and less proliferative clones in our EPC2-hTERT model (S2 Table).

Other than the standard PCA and UMAP dimensionality reduction described above, we also implemented GLM-PCA (Generalized Linear Model Principal Component Analysis) and t-SNE (t-Distributed Stochastic Neighbor Embedding) (S5C and S5D Fig). We used GLM-PCA from the glmpca package, since it offers a way to perform dimensionality reduction that is more attuned to the characteristics of sparse count data, which could potentially lead to more reliable and interpretable results than traditional PCA [71]. We also ran t-SNE using the RunTSNE function from Seaurat. We ran distinct perplexity Parameters (10, 100, 200, 300 and 400) to gain a more comprehensive understanding of the high-dimensional data structure.

## Single-cell RNA sequencing of the human esophageal epithelium

scRNA-seq datasets of healthy human esophageal epithelium [51–54] and patients with ESCC [54,72] were previously published. Before merging the datasets using Seurat V4 [65], each was subjected to a series of standard filtering processes. Initially, cells that were poorly sequenced and had a low gene count were excluded. We also identified and removed potential doublets, indicated by an unusually high gene count. Cells with a high mitochondrial gene percentage, often a marker of low quality or dying cells, were also excluded from our analysis. To enhance the accuracy of our doublet removal process, we employed scDblFinder [66], a specialized tool

for detecting doublets. Each dataset underwent independent normalization using the NormalizeData function.

We individually clustered each dataset using FindNeighbors and FindClusters. These clustered were then labeled based on known cell type markers (basal (*DST*, *KRT15*), suprabasal dividing (*MKI67*, *TOP2A*), suprabasal (*DSC2*, *DSP*), intermediate (*SERPINB3*, *PADI1*, *TGM3*), and superficial (*MT1G*, *RNASE7*, *KRT16*)) [53]. We then combined the datasets for the healthy esophageal epithelium, using the merge function in Seurat. The merged datasets were then normalized with NormalizeData, identified variable features using FindVariableFeatures (based on the 2,000 most variable features), ScaleData, and then ran PCA with RunPCA. Batch effects were significantly improved by aligning cell clusters in a shared PCA space based on the samples' respective patient identification using RunHarmony [73]. Subsequently, UMAP dimensionality reduction was applied utilizing the embeddings generated by harmony and top 20 principal components. Finally, we utilized the UCell package to compute single-cell signature scores for the highly proliferative gene signature set [74].

## Single-cell RNA sequencing cell cycle analysis

We used the 'tricycle' package in R to determine cell cycle states within a Seurat object. We utilize the Runtricycle function, specifying parameters such as slot (normalized RNA counts) and species (human). Following this, we extracted relevant data for visualization using the FetchData function, which pulled UMAP coordinates and tricycle-generated cell cycle positions. For visual representation, we created a plot of the UMAP embeddings colored by cell cycle position using plot_emb_circle_scale, which effectively displayed the distribution of cells across different cell cycle stages. Cell cycle embeddings utilized the internal reference from the tricycle package to estimate the cell cycle position, which ranged from 0 to 2pi. For ease of interpretation, we approximated the cell cycle stages with 0.5pi marking the onset of the S stage, pi representing the G2M stage, 1.5pi indicating the mid-M stage, and the range from 1.75pi to 0.25pi indicated the G1/G0 stage.

To regress out the cell cycle we defined specific gene sets for the S-phase and the G2/M-phase based on gene expression states previously identified in the literature [45]. Utilizing Seurat's CellCycleScoring function, we assigned scores for each cell based on the expression of these gene sets, effectively identifying their cell cycle phases. We then mitigated the impact of cell cycle variations on gene expression, by employing the ScaleData function within Seurat. This critical step involved regressing out the scores for the S-phase and G2/M-phase from the dataset, ensuring that all features (genes) in the Seurat object were adjusted accordingly. By doing so, we ensured that our gene expression analysis reflected biological changes beyond just cell cycle effects.

## DNA barcode ladders for spike-ins

We used spike-ins of cells, consisting of 1000, 500, and 50 cells with known barcode sequences (S4D Fig) [32]. Spike-ins were added to each sample during library preparation. The relative abundance of transcripts was quantified by comparing transcript counts to spike-in counts, allowing us to estimate the number of cells from barcode transcript counts.

## Design gRNA plasmids

We designed CRISPR RNA (crRNA) in Benchling as a 23-nt sequence using variable protospacer adjacent motifs (PAMs) 5′-TTTV (TTTA, TTTC, or TTTG). 3 crRNA targeting functional domains were designed per gene. We used the AsCas12a crRNA expression vector (pRG212, Addgene_149722) to clone the guide RNA (gRNA). We digested and linearized the pRG212

vector with BsmBI (NEB, R0580), and the sense and anti-sense crRNA oligos were annealed and phosphorylated with T4 PNK (NEB, M0201). The linearized pRG212 vector and crRNA oligo were then ligated with T4 DNA Ligase (M0202).

## Generation and validation of Cas12 expressing cell line

We obtained the AsCas12a crRNA expression vector directly from Addgene (pRG232, Addgene_149723) [75]. The plasmid was packaged and transduced into HEK293T as described in the "Lentiviral transfection" section. Two days after transduction we passaged the cells to two 10cm plates. The cells containing the Cas12-pRG232 insertion were selected with 0.5ug/mL puromycin (Takara, 631305) for three days. We determined the efficiency of the Cas12-pRG232 in our cell line by a competition-based proliferation assay. We generated two gRNAs linked to a GFP reporter; a positive control that targeted an essential gene (*PCNA*) and a non-targeting negative control (Rosa26) [75]. Cells were then transduced with either guide at 50% infection efficiency and the GFP-positive population was monitored using the BD Accuri C6 Plus Flow Cytometer every two days. We quantified editing efficiency by dividing the final GFP% timepoint by the initial GFP% timepoint per guide. We found that gRNA-Rosa26 stabilized around 98% and gRNA-*PCNA* dropped to below 1%.

## CRISPR Cas12 clonal experiment

We transduced the EPC2-hTERT with CRISPR Cas12 as described in section "*Generation and validation of Cas12 expressing cell line*". Cells were treated with puromycin (Takara, 631305) for three days. Immediately after, the cells were transduced with one of four gRNA linked to a GFP reporter: one for DKK1 exon 1, two for DKK1 exon 2 and one non-targeting negative control (Rosa26), see S3 Table. After 3 days, the cells were sorted to isolate the GFP positive cells that were successfully transduced with the gRNA. Nine days after cells were transduced with gRNA (this includes the GFP-positive sort), the cells were plated at 50 cells per well in a 6-well plate (3 biological replicates each with 6 technical replicates). A subset of cells were taken from each gRNA and were Sanger sequenced twice (forward and reverse), each gRNA reached a minimum of 70% editing efficiency. After 8 days the cells were fixed and imaged using a 10× objective on a Nikon Eclipse Ti2 fluorescence microscope.

## Small molecule inhibitors and recombinant proteins

PI3K inhibitor (GDC-0941, Cayman, 11600–10) was reconstituted at 10mM in DMSO for the stock solution. Final concentration for PI3K inhibitor treatment was 2µM. Human recombinant DKK1 (rhDkk, R&D Systems, 5439-DK) was reconstituted at 100µg/mL in 0.1% bovine serum albumin (BSA; Sigma-Aldrich, A7906). Final concentrations for rhDkk treatment were 200 ng/mL and 500 ng/mL.

## PI3Ki and rhDKK1 clonal experiments

We plated cells in 6-well plates with 50 cells per well. Cells were allowed 12 hours to properly adhere to the plate. We then treated with PI3Ki (2µM) (2 technical replicates) or rhDKK1 (100µg/mL) (3 technical replicates per dosage) and maintained treatment throughout 8 days. Similarly, DMSO was added to our control wells in the appropriate quantity. Corresponding treatment media were changed every 4 days. Cells were fixed on day 8, we first removed the media and rinsed the wells with DPBS, followed by a 10-minute treatment with 4% formaldehyde. Subsequently, the formaldehyde was aspirated away, and the wells were washed twice with DPBS. After adding 2 mL of DPBS to each well, we stained the cells with DAPI. Imaging

of the wells was then performed using a 10× objective on a Nikon Eclipse Ti2 fluorescence microscope.

## Supporting information

**S1 Fig. Clonal dynamics and growth patterns in epithelial cell growth.** (**A**) Experimental design for clonal isolation and characterization. Single EPC2-hTERT cells were plated in individual wells of a 96-well plate, and their growth was monitored over 14 days. Clones filling or nearly filling the well within this period were classified as highly proliferative, while those that did not were labeled as less proliferative. (**B**) Stacked bar plot of ratios depicts the proportion of highly proliferative and less proliferative clones (n = 6 biological replicates). (**C**) Schematic of replating experiment following the initial characterization, clones identified as highly proliferative or less proliferative were dissociated into single cells and replated into new 96-well plates. These replated cells were then allowed to grow for another 14-day period to reassess their proliferative status. (**D**) Bar plot of the ratio between highly proliferative and less proliferative subclones as described in S1C Fig (n = 3 biological replicates).
(TIF)

**S2 Fig. Bimodality quantification analysis on clone size data across time and replicates.** (**A-B**) Histograms showing the number of cell divisions for each clone based on the number of cells observed. Each set column corresponds to a separate biological replicate. Histograms are fitted with two Gaussians using a Gaussian Mixture Model (GMM). Each histogram is paired with a bar plot on the right, depicting the corresponding Bimodality Coefficient. The dashed gray lines indicate the 5/9 threshold (suggestive of a likely bimodal distribution) and the dashed black lines indicate the 1/3 threshold (suggestive of a unimodal distribution).
(TIF)

**S3 Fig. Mathematical models for epithelial cell growth.** (**A**) Bar plot showing cell divisions per clone over time (data from Fig 1C), with highly proliferative (blue) and less proliferative (green) cells. A solid gray line highlights exponential growth for comparison. (**B**) Scatter plot of the optimal parameters for the Gillespie simulations for each time point, the time point is labeled at each point. The parameters for highly proliferative and less proliferative clones are in blue and green respectively. (**C**) Schematic of growth models for the highly proliferative and less proliferative clones, with highly proliferative clones fitted using an exponential growth model and less proliferative clones requiring a logistic growth model for accurate data representation. (**D**) Scatter plot of highly proliferative and less proliferative data fitted by their respective growth rate models shown in panel C. (**E**) Live-cell imaging data from mCherry-labeled EPC2-hTERT cells over 11 days. Line plot shows the number of cells per clone as measured at 24-hour intervals. Highly proliferative clones (the top 10% largest clones) are depicted in blue, while less proliferative cells are shown in green (n = 2 technical replicates, combined). (**F**) Live-cell imaging data from phase imaging of unlabeled EPC2-hTERT cells over 11 days. Line plots of normalized clonal area ($\mu m^2$) over time, measured at 12-hour intervals, for EPC2-hTERT cells (n = 2 technical replicates, combined).
(TIF)

**S4 Fig. Supporting analysis for heritability and barcoding experiments in Fig 3.** (**A-B**) Scatter plot depicting the number of cells in sample A and B for each barcode captured by sequencing. The Pearson correlations were 0.511 and 0.373 with p-values of 4.27e-10 and 1.17e-5 in the two replicates shown in A and B. The embedded box plots summarize the findings comparing simulated data, our experimental replicates, and spike-in positive controls that should show high correlation (n = 2 technical replicates). (**C**) Histograms showing the Pearson

correlations for simulated data (see methods section: Heritability analysis), experimental data, and spike-ins positive controls. Both replicates, shown in blue, fall outside the simulation ranges, indicating memory of epithelial cell proliferative capacity. (**D**) Scatter plot showing data from spike-ins included in the experiment with the number of cells versus the number of reads per million for the barcode captured by sequencing. (**E-F**) Bar plots of rank-ordered barcodes, representing barcode distribution after 8 days of growth. (**G**) Box plot of percentage of recovered barcoded cells at 500,1000, 5000 and 10000 cells per well, detailed in methods "Clonal density experiments" (n = 2 technical replicates). (**H**) Rank-ordered barcodes grouped by density while the heat bar displays the log(Reads per Million). (**I**) Box plot of data from Fig 3D showing the number of mCherry labeled cells per clone with the data split by initial seeding density (n = 2 biological replicates each with 6 technical replicates). (**J**) Box plot showing quantification of the initial seeding density for the experiment in Fig 3D to confirm that each condition started with the same number of mCherry labeled cells. The plot depicts the number of mCherry clones measured at day 8.
(TIF)

**S5 Fig. Molecular characterization of highly proliferative and less proliferative clones. (A**) UMAP plots with examples of lineages from less proliferative clones. (**B**) UMAP plots with examples of lineages from highly proliferative clones. (**C-D**) Dimensionality reduction plots of scRNA-seq data. These plots utilize different dimensionality reduction methods (including UMAP, PCA, GLMPCA, and t-SNE with varying perplexity settings of 10, 100, 200, 300, and 400) to visualize scRNA-seq data. Cells are color-coded to distinguish between highly proliferative and less proliferative states, which we assigned based on barcodes. (**E**) UMAP projection of scRNA-seq data with cells colored based on their cell-cycle position using a circular color scale. Discrete stage labels are placed in approximate positions on the circular legend.
(TIF)

**S6 Fig. Clustering and performance metrics for scRNA-seq and quantification of markers Ki67 and KRT14. (A**) UMAP plot with regressed out of cell cycle Influence, cells are labeled according to Louvain clusters. (**B**) UMAP and PCA plots of cells processed through two different 10x Genomics lanes illustrate the consistency and distribution of the cells across the lanes. (**C**) Barplot showing the percentage of cells in each state split by lane 1 and lane 2. (**D**) Scatter plot of the number of cells per clone against *MKI67* (left) and *KRT14* (right) expression from the EPC2-hTERT scRNA-seq dataset. (**E**) Immunofluorescence data in Fig 5 staining for Ki67 and KRT14 in clones of different sizes. Scatter plot shows Ki-67 and KRT14 intensities with each cell labeled according to the total number of cells in its respective clone (n = 2 biological replicate and n = 2 technical replicates).
(TIF)

**S7 Fig. Perturbations to WNT and PI3K can alter clonal growth dynamics. (A**) Experiment testing the effects of PI3K inhibitor on the growth of clones over 8 days. Box plot shows the number of cells per clone with and without PI3K inhibitor (iPI3K) (n = 2 biological replicates and n = 2 technical replicates) (p-value equal to 4.33e-5 and 1.08e-9 respectively). (**B**) Experiment testing the effects of recombinant DKK1 protein on growth of clones over 8 days. Box plot shows the number of cells per clone at two different concentrations of human recombinant DKK1 (n = 2 biological replicates and n = 3 technical replicates per dosage). (**C**) Outline of experimental design for the CRISPR-Cas12a targeted knockout (KO) used to investigate the role of DKK1 in clone expansion. (**D**) Box plot showing the number of cells per clone split by gRNA targeting Rosa26 and DKK1 (n = 3 biological replicates) (p-value equal to 1.81e-1, 5.54e-12 and 1.06e-2 respectively). (**E**) Box plot showing the percentage of highly proliferative

clones per well split by gRNA targeting Rosa26 and DKK1 (n = 3 biological replicates each with 6 technical replicates) (p-value equal to 2.17e-01, 4.16e-02 and 6.90e-05 respectively). We selected cutoff size for a highly proliferative clone by taking the top 10% of the average of the Rosa26 control data. We then used that cutoff size to determine which clones are highly proliferative in the DKK1 KO data.
(TIF)

**S8 Fig. The healthy human esophageal epithelium contains rare basal cells with a similar profile to the highly proliferative clones found in vitro.** (**A**) A schematic of the integration of numerous published scRNA-seq datasets for healthy esophageal epithelium comprising 24 patients and 62,216 cells. (**B**) UMAP plot showing each cell labeled by its dataset after batch correction. (**C**) UMAP plot in which the color of each cell shows their predicted cell-cycle position using a circular color scale. Discrete stage labels are placed in approximate positions on the circular legend.
(TIF)

**S1 Table. Numbers from clonal isolation and replating experiments.**
(XLSX)

**S2 Table. Top 50 differentially expressed genes identified between highly and less proliferative clusters.**
(XLSX)

**S3 Table. sgRNA for CRISPR Cas12.**
(XLSX)

## Acknowledgments

We thank all members of the Shaffer lab for feedback on experiments and the manuscript, A. Singh for thoughtful discussions and ideas, G. Harmange for help on developing image processing and barcoding pipelines, and A. Muir for providing cell lines.

## Author Contributions

**Conceptualization:** Raúl A. Reyes Hueros, Rodrigo A. Gier, Sydney M. Shaffer.

**Data curation:** Raúl A. Reyes Hueros, Rodrigo A. Gier.

**Formal analysis:** Raúl A. Reyes Hueros.

**Funding acquisition:** Sydney M. Shaffer.

**Investigation:** Raúl A. Reyes Hueros, Rodrigo A. Gier.

**Methodology:** Raúl A. Reyes Hueros, Rodrigo A. Gier, Sydney M. Shaffer.

**Project administration:** Raúl A. Reyes Hueros, Sydney M. Shaffer.

**Resources:** Sydney M. Shaffer.

**Software:** Raúl A. Reyes Hueros, Rodrigo A. Gier.

**Supervision:** Sydney M. Shaffer.

**Validation:** Raúl A. Reyes Hueros, Sydney M. Shaffer.

**Visualization:** Raúl A. Reyes Hueros.

**Writing – original draft:** Raúl A. Reyes Hueros, Sydney M. Shaffer.

**Writing – review & editing:** Raúl A. Reyes Hueros, Rodrigo A. Gier, Sydney M. Shaffer.

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
