## [Decision Letter · Decision Letter 0]

8 Sep 2023

Dear Dr. Shaffer,

Thank you very much for submitting your manuscript "Non-genetic differences underlie variability in proliferation among esophageal epithelial clones" for consideration at PLOS Computational Biology.

As with all papers reviewed by the journal, your manuscript was reviewed by members of the editorial board and by several independent reviewers. In light of the reviews (below this email), we would like to invite the resubmission of a significantly-revised version that takes into account the reviewers' comments.

In particular, we suggest to: emphasize the novelty of your studies, especially for what concerns the density effect; give more relevance to the conclusions you can draw from the mathematical model;  make the manuscript more consistent as suggested by the reviewers, whose advices and criticisms should be addressed.

We cannot make any decision about publication until we have seen the revised manuscript and your response to the reviewers' comments. Your revised manuscript is also likely to be sent to reviewers for further evaluation.

Sincerely,

Andrea Ciliberto

Academic Editor

PLOS Computational Biology

Stacey Finley

Section Editor

PLOS Computational Biology

Reviewer's Responses to Questions

**Comments to the Authors:**

Reviewer #1: The manuscript, “Non-genetic differences underlie variability in proliferation among esophageal epithelial clones”, describes a phenomenon where proliferation capacity of clones are inherited over cell passages through non-genetic means. This is not the first study of its kind but it brings some novel approaches such as barcoding into the mix. The authors present some elegant ways to decipher between multiple hypotheses. There are some good validation experiments performed perturbing certain signaling pathway proteins. However, we are not sure why this is in PLOS Comp Bio as it has very little computational biology presented (only simple 1 or 2 states models presented). Overall we think it is an interesting study.

Major comment

• Make sure all panels in all figures should be mentioned in the main text.

• The tables mentioned in text cannot be found.

• The methods are very general and hard to follow from the main text. Please make sure methods are detailed enough to enable readers to repeat the experiments and simulations.

• Sample size, both number of biological and of technical replicates, need to be reported.

• Figure1: Individual epithelial clones have unique capacities for growth and differentiation

o The authors used top 10% of clones ranked by size as a criterion to assign whether a clone is expanding or committed. However, it is unclear how this threshold was selected. The authors did not mention other criteria thresholding in other sections of the paper, but there is at least one place where the percentage of expanding colonies is changed (Fig S4C), which differs from the definition of expanding clones as the top 10% in size and suggests that the authors used other thresholds in that experiment.

o The authors assumed that each clone from a cell line is genetically identical. This is probably not the case as cell lines can generate genetic variation very quickly in a few generations. The genetic similarity amongst clones can be tested by small sample bulk sequencing.

o Fig S1A: the author should also test if cells picked from the committed clone can also give rise to both committed and expanding clones. It was only tested that the expanding clone can give rise to the committed.

• Figure 2: Cell states underlying proliferative potential are heritable through cell division

o Fig 2B and S2A: even though the p-values are low, the correlation coefficients of 0.511 and 0.373 should not be considered high, and the data points seem to have high variation. The 4 axes from the two plots show very different range of cell numbers. Again, the number of experimental replicates is not stated clearly in main text, and a reason for plotting the results in separate plots are not clarified. It is not convincing enough to draw the conclusion that the proliferation capacity between splits A and B are correlated.

o Furthermore, since it has already been shown the impact of cell density in the next section, there may be variation in cell density in the two split plates. Have the authors consider growing the individual split cells in separate wells? This way, the density variable can be removed (actually we are not sure why this is not the experimental design to begin with as it is a more straight forward experiment)

o A time period of 4 doublings is used. Can the authors explain the choice of this interval. Will the correlation change if the interval varies?

• Figure 3: The transcriptional state of expanding and committed cells are distinct

o Fig 3D: no clear clusters are formed between the two states. Although the expanding cells locate more in the inner circle on the UMAP, that may be due to batch effects. The authors have not mentioned the number of batches/replicates, and the experiment should be repeated.

o “We then labeled cells based on their respective clone sizes. Consistent with our previous observations, cells in the top 10% of clone sizes were labeled as expanding, while those in the remaining smaller clones were labeled as committed.” The statement sounds cyclic.

o Fig S4C: need to explain how the expanding colonies are assigned in a quantitative way. See comment above

• Figure 4: A rare subset of cells with an expanding transcriptional signature was found in the human esophageal epithelium

o Fig 4 D and H: the author should clearly state how the expanding signature is selected.

o There are more expanding cells in cancer compared to pre-cancer. This is hardly a noel observation and I do not know what the authors are trying to get at with this human figure. It does not really speak to the inheritability of these states.

Minor comment

• Please make sure all figure labels are consistent between the text and actual figures. Too many figures mentioned in text do not match the actual figure panels.

• Reasoning and consistency in units should be specified. For example, Fig2D used ‘number of cell divisions’ in the plot but used number of cells per clone in text. There is no clarification how the two units are measured/converted.

• Color used for each experiment condition should be consistent (see fig S4 B and C)

Reviewer #2: Comments to the authors:

The manuscript entitled “Non-genetic differences underlie variability in proliferation among oesophageal epithelial clones” by RA Reyes Hueros et al attempts to provide insight into the molecular mechanism underlying variable proliferation rates for individual clones found in an in-vitro model of oesophageal epithelial cells (EPC2-hTERT) and then aims to find biological relevance by directly comparing the in-vitro obtained data with healthy oesophageal tissue and oesophageal squamous cell carcinoma (ESCC) patient samples.

I would like to start by mentioning that, to the eyes of this reviewer, the observations, experiments and molecular biology concepts described in the manuscript are not novel but rather a re-telling of old school “cell line” cloning observations with a spin of using state-of-the-art tools. It is in that view that, the apparent “novelty” here described has been observed in numerous cases and quite probably is a general phenomenon found in cellular cloning experimentation. The variable rate of cloning efficacy for any given cell line has been well documented and it has been ascribed to numerous cellular and molecular features (cellular crowding, condition media, cell-to-cell communication, cell cycle phase, etc) and by no means is a particular feature of the cell model tested here.

Major comments:

1- There is no description on how the EPC2-hTERT original cell line was generated or how the hTERT was introduced or controlled to increase its expression. Moreover, and extremely important for this kind of studies… was the original cell line already cloned or is it a heterogeneous batch?

2- How genetically stable is the EPC2-hTERT cell line? If it is not stable… for how many passages can the non-genetic compartment be interrogated?

3- Schemes depicting potential experimental outcome must not be part of figures as they already suggest the results of the experiment prior to the actual evaluation of the data. Moreover, they suggest the possibility that the author contemplated all possible scenarios whereas the authors may be mistaken in their observation and analysis. Please revise all figures depicting schemes for data outcome.

4- As in 3-, though the authors in the text contemplate a “minimalistic” model of proliferation they are leaving out multiple possibilities such as cell death (any kind), senescence, cell cycle progression speed among others. Even if the model turns out to be the right one, possibilities must be explored and discussed.

5- In the first bit of the manuscript (Figure 1), cells are fixed and imaged. Therefore, the clones analysed on every time frame are different. I suggest the authors to re-run this experiment but this time following the same clones over time by live imaging. In this way, many additional layers of information may be also extracted from the data and used to complement the model.

6- Also, how many times the plating and counting has been made for each data point? How many technical and how many biological replicates? Which is the variance between plates? Is the count based on cell counting or colony size? Is there any kind of filter used to remove small colonies? Are small colonies taken into account at all?

7- Which is the number of cells or range that the authors used to define highly proliferative from non-proliferative? Is this arbitrary or was model based on the data? If arbitrary, make it explicit in the main manuscript (the authors suggest 10% in the manuscript but further explanation is needed). Was this consistent among biological replicates analysed in the same experiment/across different experiments?

8- Do smaller (in number of cells) clones stop proliferating or their rate of proliferation is reduced? This can be also analysed by live imaging.

9- Certain definitions presented here needs to be expanded and explained in the context of the manuscript… For instance, “… We included the non-proliferative state because some clones contain cells that stop proliferating and show morphological features consistent with differentiation…”. Differentiation into? Also, a change in morphology does not necessarily mean “differentiation”. What do the authors imply here? I suggest the authors to clarify the concept.

10- Figure 1c is not clear and the legend does not help either.

11- I think that definitions here are needed. For instance, “… Thus, we separated the clone size data (Fig. 1C) into the top 10%, which we named "expanding". For the remaining clones, we assumed that they followed a pattern consistent with the halting of proliferation, so we named them "committed"…” . Committed to what? The wording here needs to be revised as the word “committed”, and depending on the field of research, alludes to multiple biological scenarios (determination, differentiation, lineage, etc).

12- In the experiment of the 96-well plate… How many clones were plated? How many clones survived? How many “stop” proliferating after several cell divisions? How many did not grow at all? How many were retrieved? How many wells were empty after the sorting? How many times the experiment was carried out? What happens if you re-plate the clones that displayed halted proliferation? All this data cannot be neglected as all these potential phenotypes contribute to the overall outcome of the population.

13- In the following paragraph, “… If genetic differences in these clones were responsible for the growth differences, we would expect that all of the subclones derived from the highly proliferative parent clone would grow similarly to the parent. Instead, we found that the subclones followed a similar growth distribution to the initial population (Fig. S1A). Thus, we can conclude that the highly proliferative clones are not a genetically distinct subset of the EPC2-hTERT models…” the authors suggest that the observed phenotype is non-genetic as it re-shuffles to the original proliferation distribution observed in the population. However, and as stated above regarding the novelty of the study, this feature is quite probably a general phenomenon found in cellular cloning that can stem from a multitude of biological inputs. I suggest the authors to tone this down.

14- In Figure 2… it is my opinion that the data depicted here directly contrasts the data shown in S1A. The authors suggest a “… When we compared the number of reads in split A and split B, we found a significant correlation in the number of reads for each barcode across the different plates (Fig. 2B)…”. However, if the highly proliferating clones, once plated at single cell confluence re-constitutes the initial grow heterogeneity it should be expected a 90% “committed” (not the right wording here) and 10% expanding… how does the authors explain the high correlation observed? The barcodes, used as proxy for cell proliferation here, should also display the same behaviour. I suggest the authors to look deeper into the “barcodes” dataset as to this reviewers’ view, both experiments should show, if the authors are right, a degree of concordance.

15- Also, in Legend to figures, especially in Supplementary Figures, there is a lot of description and interpretation of the results. This must be removed from legends to figures. Legends should only describe the data depicted and the way the data is represented.

16- Also, in relation to data and Figure 2, it would be useful to describe how many barcodes where detected, how much data was removed, how many cells are estimated to be properly labelled (1 copy per genome), etc. The inclusion of those metrics will help the potential reader to better evaluate the dataset.

17- In Figure 2b. It reads “Barcodes show memory of proliferative capacity”. Barcodes are not showing anything… The cells expressing/labelled with a given barcode suggests “something”.

18- In Figure 2c the authors attempt to explain the effect of cell number in the proliferation capability for clonal growth. Once again, this is not novel and it has been shown endless times. The more cells you have on a plate whilst not in confluence conditions, the better cells grow and proliferate.

19- Regarding the single cell experiments depicted in Figure 3. It is not clear to this reviewer how the experiment was performed. The figure suggest that the clones are collected based on the number of cells (e.g CellCelector?) and then pooled? Or is it just clonal growth by dilution and then the plate is collected and processed for single cell analysis? The distinction is fundamental as, based on the results shown here (e.g. Fig. 3E), many more cells from the expanding clones should be sequenced when compared to the “committed lineage” found in the plate which would be minimal in the contribution to the overall pool. By analysing the UMAP depicted in Fig. 3D this does not seem to be the case and, moreover, it looks like there are more “committed” cells than proliferation ones. This reviewer may be missing something here, but the data depicted is telling a contrasting story. Could the authors please develop?

20- Also, and given the data displayed in Figure S1A, phenotype interconversion is happening. It would be informative if the authors would explore the data for cells that belong to a given “lineage” that are actually changing their phenotypic output. Please provide numbers and discuss possibilities.

21- The authors are measuring “steady-state RNA levels” and not “active transcription”… Therefore, the authors should not argue that they are observing “transcriptional differences” but rather “gene expression levels differences”. The difference is not un-important.

22- Information about cell cycle embedding for the single cell analysis is not available within the manuscript. How was this performed? How many gene expression markers were used? What is the underlying theory behind the colour gradient code? I have serious doubts about inferring cell cycle transitions using gene expression profiles… cell cycle phases yes… cell cycle transitions… not sure at all. Please develop the framework.

23- Regarding the gene expression signature associated to proliferative clones found in the in-vitro system and then tested with datasets derived from healthy tissue and patient samples… More information must accompany the analysis of these datasets. Is the data batch corrected somehow? is the data “integrated” somehow? If so, which are the anchors and by which method was this achieved.

24- Regarding my previous comment… at least 1/3 (perhaps a little bit more) of the ESCC patients do not follow the signature. Do the authors have access to the clinical features from each sample? Perhaps the signature can be further refined. Also, Fig. 4A shows 24x Patients and I count 26 bars in the histogram (healthy? Or tissue surrounding ESCC?). Similarly, Fig. 4E states 55x Patients… I count 64 bars in the histogram. Also, it would be interesting to include examples of patient samples from both extremes and show them in contrast to healthy samples. A few UMAPs would be informative here.

25- Also in relation to my previous point, I’ve just roughly calculated the number of cells per patient used in the previous analysis and I’ve found that for healthy tissue you’d get ~3000 cells per sample (patient) whereas in the ESCC panel you’d only get ~441 cells per sample (patient). Will that be sufficient to support the conclusions here drawn? It would be informative to explore the integration of both datasets Healthy vs ESCC. I must stress that I’m a little sceptical about the validity of this comparison given the divergence in the number of cells in both datasets. Please develop and include metrics for all this.

26- Looking at the data suggesting that markers of highly proliferating cells can be derived from the in vitro system, this reviewer keeps wondering whether the signature is nothing else but a proliferation signature that can be found pretty much in every proliferating cell line? How many markers are used to define highly proliferating cell from healthy and ESCC patients? I find that, defining a population by the expression of a few transcripts, especially from tissue samples, may be misleading. This information/concepts must be expanded in main manuscript.

27- The finding that WNT and PI3K signalling pathways are altered in highly proliferating cells is not surprising as it has been reported numerous times. Therefore, this reviewer finds the experiments of CRISPR KO and PI3K inhibitors not informative enough to make a step forward into describing the molecular mechanism. Also, upregulation/downregulation of a transcript does not necessarily mean enhanced translation of the corresponding protein. Therefore, I suggest the reviewers to tone down all claims directly linking transcript modulation with the presence/absence of a protein/pathway.

28- The discussion must be toned down as to describe what the manuscript is actually showing.

Minor comments:

1- The manuscript must be revised for consistency. The wording must reflect the actual data.

2- Figure legends must be re-shuffled to not include explanatory text.

3- The methods section must be further developed to include every single experiment and analysis so any given reader can reproduce the results here presented.

4- This may be a consequence of file conversion but, Figures are fuzzy. Also, please work on the figures to make them auto explicative.

5- A data analysis section must be included. Mentioning that Seurat was used to combine the datasets is hardly enough. Metrics should be given.

6- In Supplementary Fig.5 the cells labelled as expanding in the UMAP seems to be bigger (as in bigger dot) than the ones in the markers UMAPs. This representation seems a little misleading to this reviewer. Please make it uniform.

Reviewer #3: This is an interesting manuscript that address the important question of the contribution of non-genetic mechanisms to differences in growth potential. By combining scRNA-seq with barcoding in an in vitro model of esophageal epithelium, they show that non-genetic proliferative capacity in a heritable trait that can be influenced by extrinsic factors such as cell density. Focusing on highly proliferative clones they found a cellular state which is associated with upregulation of the WNT and PI3K pathways. Finally, they show that this proliferative cellular state can also be found to some extant in patient data.

I think this manuscript address a fundamental largely overlooked aspect of non-genetic variation however there are several points that needs to be addressed to make the claims stronger, namely:

1) The author propose an that what they findings can not be explained by “a simple two-parameter model of self-renewal and differentiation” however throughout the manuscript the two main phenotypes are proliferative and arrested. If the authors could show that the smaller colonies are showing distinct markers of differentiation, I think that could significantly contribute to their work.

2) I would be good to calculate what are the transition rates between the two states defined above.

3) The authors note that the proliferative state is heritable though a “limited number of cell division” but don’t show what this limit is. It would be good to include data that support this claim and shows that the correlation decays with time.

4) The observation that density impacts growth kinetics is important but very well known. It would be good if the authors could clarify how the information they are providing is new compared to what is known in the field.

5) My biggest concern is with regards to expansion signature. The authors define a transcriptional state that is associated with expanded clones. This state is associated with Pi3K and WNT. It would be good to corroborate that this signature doesn’t disappear with different cell cycle regression methods as well is to show how general it is. For instance, is this specific to esophageal epithelium or a more general cellular trait.

**Have the authors made all data and (if applicable) computational code underlying the findings in their manuscript fully available?**

Reviewer #1: Yes

Reviewer #2: **No: **The authors state "All code and data used will be made available upon publication"

Reviewer #3: Yes

PLOS authors have the option to publish the peer review history of their article (what does this mean?). If published, this will include your full peer review and any attached files.

Reviewer #1: No

Reviewer #2: No

Reviewer #3: **Yes: **Yaara Oren
---

## [Decision Letter · Decision Letter 1]

24 Apr 2024

Dear Dr. Shaffer,

Thank you very much for submitting your manuscript "Non-genetic differences underlie variability in proliferation among esophageal epithelial clones" for consideration at PLOS Computational Biology. As with all papers reviewed by the journal, your manuscript was reviewed by members of the editorial board and by several independent reviewers. The reviewers appreciated the attention to an important topic. Based on the reviews, we are likely to accept this manuscript for publication, providing that you modify the manuscript according to the review recommendations.

In particular, one relevant point raised by the reviewers concerns the number of replicates. The authors should specify that the scRNA-seq was performed once. For all other experiments at least 2 biological replicates are needed. This is especially true for Fig S5, 5C, S6B, S6C, S6D and S6F which apparently only have technical replicates. The authors should also address the two issues raised by reviewer 1 (regarding the definition of the threshold based on live cell imaging data,  and capture effectiveness of small clones). Comments 4 and 10 of reviewer 2 should also be addressed.

Sincerely,

Andrea Ciliberto

Academic Editor

PLOS Computational Biology

Stacey Finley

Section Editor

PLOS Computational Biology

Reviewer's Responses to Questions

**Comments to the Authors:**

Reviewer #1: Just some additional explanations:

The authors gave clear explanation how the 10% threshold is selected based on experimental observations. However, it is only mentioned that “A small fraction of clones expanded to fill entire wells, while the majority exhibited reduced 100 growth or stopped growing entirely”, or in the 6-well plates/live cell imaging condition, the authors mentioned the presence of two types of clones. Yet, the authors did not give enough evidence in how the clone classification were binarized. It is possible that the clone size is a range of values, not necessarily following a bimodal distribution. Especially in the live cell imaging figures (Fig. S2 E and F), the change from the green to the blue lines can be perceived as gradual. The authors need further statistical tests/ plots to supports the assumption of a bimodal distribution of clone sizes.

Though the authors argued that small clones can be more effectively recovered when using 6-well plates, it is not obvious to the reviewer why the size of the plates reduce RNA-seq capture effectiveness of small clones. Thus, the authors should provide further explanation or data to support the statement.

Reviewer #2: In this revised manuscript by Reyes Hueros et al the authors attempt to complement their initial submission with additional insights. It is the strong opinion of this reviewer that all the data requested by the reviewers and displayed on “data for the reviewers” should be included as an integral part of the manuscript for everyone to see and evaluate. Of course, this would include a major reshuffling as major data bits should be included in main Figures and therefore fully explained in the main manuscript (e.g. phenotype reversibility).

Major comments:

1- Regarding the actual data. Although I appreciate the effort in the data analysis and observations to dismiss the effect of genetic alterations on the phenotype, I think that at least a whole exome sequencing (WES) of the initial “parental” population should accompany the manuscript. This will undoubtfully sway away (minimize to a certain extent) doubts about the genetic compartment playing a role in the phenotypic divergence. Alternatively, though not my preferred approach, mutation calling could be extracted from the single cell data. Please consider both options.

2- Regarding the “less proliferative clones” not been able to revert to “highly proliferative”… Did the authors try to stain for b-galactosidase? It would be interesting to see whether the non-proliferating clones or the clones that are transitioning from proliferative to less proliferative become senescent. This observation could improve the model and could provide quite relevant biological data.

3- The authors need to revise their manuscript to tone down many of their claims. For instance… “Clonal growth dynamics are largely heritable through 4 cell doublings…”. I do not think the data supports “largely”. Also, the authors mention in their Summary and Introduction: “tissue regeneration” as one of the aims encompassing their research. There is no evidence whatsoever in the manuscript to advance/suggest this kind of claims. Please remove.

4- What is Supplementary Figure 6A trying to show? Is it that both phenotypes express equivalent levels of the reporter? If not… is it the translocation of the reporter? If the latter, I cannot see any difference in nuclear vs cytoplasmic staining. Please include a figure where the difference in nuclear vs cytoplasmic staining for the reporter is clearly visible. Also, is the data analysis here done using confocal microscopy? Or regular fluorescence microscopy? By the description in M&M is just fluorescence microscopy with all the quantification caveats associated to it (e.g fluorescence bleeding, quality of lenses filters, etc). I strongly suggest the authors to re-run the analysis using confocal microscopy.

5- In the same line, in M&M in the FACS section, the authors state that “all cells have a fluorescent tag”. Please specify which cells? For which experiment? Lentiviral transduction was used across the manuscript in multiple occasions and this could problematic as for instance the Fox01 reporter is GFP tagged. Please develop.

6- Please describe in-full the usage of the Fox01 reporter within the main text.

7- M&M is plagued by data interpretation and conclusions. Please move it away from M&M and re-place all this content in the main manuscript.

8- Some figures also display data interpretation. Please remove from M&M and place in the main manuscript.

9- Please re-organize the M&M section as information about individual assays is currently scattered across multiple sections. Super hard to follow.

10- In the section Clonal isolation and replating in the M&M section there is not description about how this is done. Is FACS based? CellCellector? Is it based on GFP expression? I suggest the authors to include all the data related to single cell sorting in the form of supplementary figures for every time they show a lentiviral transduction, specify gating conditions and percentages obtained. Also display final enrichment after sorting. This data should already be available to the authors.

11- The Discussion needs to be revamped to explore state-of-the-art knowledge. The current version only describes the manuscript observations and it falls flat.

Minor comments:

1- The “Lentiviral transfection” title in the M&M section does not work as it does not reflect what is actually done.

2- Some of the “transcriptional state” are still there. Please replace for, as suggested by the authors, gene expression or something alike.

Reviewer #3: All issues were addressed

**Have the authors made all data and (if applicable) computational code underlying the findings in their manuscript fully available?**

Reviewer #1: Yes

Reviewer #2: **No: **GEO accesion numbers for sequencing data generated in the context of this manuscript are not provided

Reviewer #3: Yes

PLOS authors have the option to publish the peer review history of their article (what does this mean?). If published, this will include your full peer review and any attached files.

Reviewer #1: No

Reviewer #2: No

Reviewer #3: **Yes: **Yaara Oren

Figure Files:

Data Requirements:

Reproducibility:

References:

---

## [Editor Report · Decision Letter 2]

24 Jul 2024

Dear Dr. Shaffer,

We are pleased to inform you that your manuscript 'Non-genetic differences underlie variability in proliferation among esophageal epithelial clones' has been provisionally accepted for publication in PLOS Computational Biology.

Best regards,

Andrea Ciliberto

Academic Editor

PLOS Computational Biology

Stacey Finley

Section Editor

PLOS Computational Biology

---

## [Editor Report · Acceptance letter]

21 Oct 2024

PCOMPBIOL-D-23-01079R2 

Non-genetic differences underlie variability in proliferation among esophageal epithelial clones

Dear Dr Shaffer,

I am pleased to inform you that your manuscript has been formally accepted for publication in PLOS Computational Biology. Your manuscript is now with our production department and you will be notified of the publication date in due course.

With kind regards,

Zsuzsanna Gémesi
